# On the Near-Optimality of Local Policies in Large Cooperative Multi-Agent Reinforcement Learning

**Washim Uddin Mondal**                                          *wmondal@purdue.edu*
*School of IE and CE, Purdue University*

**Vaneet Aggarwal**                                              *vaneet@purdue.edu*
*School of IE and ECE, Purdue University*

**Satish V. Ukkusuri**                                           *sukkusur@purdue.edu*
*Lyles School of Civil Engineering, Purdue University*

**Reviewed on OpenReview:** *https://openreview.net/forum?id=t5HkgbxZp1*

## Abstract

We show that in a cooperative $N$-agent network, one can design locally executable policies for the agents such that the resulting discounted sum of average rewards (value) well approximates the optimal value computed over all (including non-local) policies. Specifically, we prove that, if $|\mathcal{X}|, |\mathcal{U}|$ denote the size of state, and action spaces of individual agents, then for sufficiently small discount factor, the approximation error is given by $\mathcal{O}(e)$ where $e \triangleq \frac{1}{\sqrt{N}} \left[ \sqrt{|\mathcal{X}|} + \sqrt{|\mathcal{U}|} \right]$. Moreover, in a special case where the reward and state transition functions are independent of the action distribution of the population, the error improves to $\mathcal{O}(e)$ where $e \triangleq \frac{1}{\sqrt{N}} \sqrt{|\mathcal{X}|}$. Finally, we also devise an algorithm to explicitly construct a local policy. With the help of our approximation results, we further establish that the constructed local policy is within $\mathcal{O}(\max\{e, \epsilon\})$ distance of the optimal policy, and the sample complexity to achieve such a local policy is $\mathcal{O}(\epsilon^{-3})$, for any $\epsilon > 0$.

## 1 Introduction

Multi-agent system (MAS) is a powerful abstraction that models many engineering and social science problems. For example, in a traffic control network, each controller at a signalised intersection can be depicted as an agent that decides the duration of red and green times at the adjacent lanes based on traffic flows (Chen et al., 2020). As flows at different neighboring intersections are inter-dependent, the decisions taken by one intersection have significant ramifications over the whole network. How can one come up with a strategy that steers this tangled mesh to a desired direction is one of the important questions in the multi-agent learning literature. Multi-agent reinforcement learning (MARL) has emerged as one of the popular solutions to this question. *Cooperative* MARL, which is the main focus of this paper, constructs a *policy* (decision rule) for each agent that maximizes the aggregate cumulative *reward* of all the agents by judiciously executing exploratory and exploitative trials. Unfortunately, the size of the joint *state-space* of the network increases exponentially with the number of agents. This severely restricts the efficacy of MARL in the large population regime.

There have been many attempts in the literature to circumvent this *curse of dimensionality*. For example, one of the approaches is to restrict the policies of each agent to be *local*. This essentially means that each agent ought to take decisions solely based on its locally observable state. In contrast, the execution of a *global* policy requires each agent to be aware of network-wide state information. Based on how these local policies are learnt, the existing MARL algorithms can be segregated into two major categories. In independent Q-learning (IQL) (Tan, 1993), the local policies of each agent are trained independently. On the other hand, in Centralised Training with Decentralised Execution (CTDE) based paradigm, the training of local policies

are done in a centralised manner (Oliehoek et al., 2008; Kraemer and Banerjee, 2016). Despite the empirical success for multiple applications (Han et al., 2021; Feriani and Hossain, 2021), no theoretical convergence guarantees have been obtained for either of these methods.

Recently, mean-field control (MFC) (Ruthotto et al., 2020) is gaining popularity as another solution paradigm to large cooperative multi-agent problems with theoretical optimality guarantees. The idea of MFC hinges on the assertion that in an infinite pool of homogeneous agents, the statistics of the behaviour of the whole population can be accurately inferred by observing only one representative agent. However, the optimal policy given by MFC is, in general, non-local. The agents, in addition to being aware of their local states, must also know the distribution of states in the whole population to execute these policies.

To summarize, on one hand, we have locally executable policies that are often empirically sound but have no theoretical guarantees. On the other hand, we have MFC-based policies with optimality guarantees but those require global information to be executed. Local executability is desirable for many practical application scenarios where collecting global information is either impossible or costly. For example, in a network where the state of the environment changes rapidly (e.g., vehicle-to-vehicle (V2V) type communications (Chen et al., 2017)), the latency to collect network-wide information might be larger than the time it takes for the environment to change. Consequently, by the time the global information are gathered, the states are likely to transition to new values, rendering the collected information obsolete. The natural question that arises in this context is whether it is possible to come up with locally executable policies with optimality guarantees. Answering this question may provide a theoretical basis for many of the empirical studies in the literature (Chu et al., 2019) that primarily use local policies as the rule book for strategic choice of actions. But, in general, how close are these policies to the optimal one? In this article, we provide an answer to this question.

## 1.1 Our Contribution

We consider a network of $N$ interacting agents with individual state, and action space of size $|\mathcal{X}|$, and $|\mathcal{U}|$ respectively. We demonstrate that, given an initial state distribution, $\boldsymbol{\mu}_0$, of the whole population, it is possible to obtain a non-stationary *locally executable* policy-sequence $\tilde{\boldsymbol{\pi}} = \{\tilde{\pi}_t\}_{t \in \{0,1,\cdots\}}$ such that the average time-discounted sum of rewards (value) generated by $\tilde{\boldsymbol{\pi}}$ closely approximates the value generated by the optimal policy-sequence, $\boldsymbol{\pi}^*_{\mathrm{MARL}}$. In fact, we prove that the approximation error is $\mathcal{O}(e)$ where $e \triangleq \frac{1}{\sqrt{N}}\left[\sqrt{|\mathcal{X}|} + \sqrt{|\mathcal{U}|}\right]$ (Theorem 1). We would like to clarify that the optimal policy-sequence, $\boldsymbol{\pi}^*_{\mathrm{MARL}}$, is, in general, *not* locally executable. In a special case where reward and transition functions are independent of the action distribution of the population, we show that the error can be improved to $\mathcal{O}(e)$ where $e \triangleq \frac{1}{\sqrt{N}}\sqrt{|\mathcal{X}|}$ (Theorem 2).

Our suggested local policy-sequence, $\tilde{\boldsymbol{\pi}}$ is built on top of the optimal mean-field policy sequence, $\boldsymbol{\pi}^*_{\mathrm{MF}}$ that maximizes the infinite-agent value function. It is worth mentioning that $\boldsymbol{\pi}^*_{\mathrm{MF}}$, in general, is not local—agents require the state-distribution of the $N$-agent network at each time-step in order to execute it. Our main contribution is to show that if each agent uses infinite-agent state distributions (which can be locally and deterministically computed if the initial distribution, $\boldsymbol{\mu}_0$ is known) as proxy for the $N$-agent distribution, then the resulting time-discounted sum of rewards is not too far off from the optimal value generated by $\boldsymbol{\pi}^*_{\mathrm{MARL}}$.

Finally, we devise a Natural Policy Gradient (NPG) based procedure (Algorithm 1) that approximately computes the optimal mean-field policy-sequence, $\boldsymbol{\pi}^*_{\mathrm{MF}}$. Subsequently, in Algorithm 2, we exhibit how the desired local policy can be extracted from $\boldsymbol{\pi}^*_{\mathrm{MF}}$. Applying the result from (Liu et al., 2020), we prove that the local policy generated from Algorithm 2 yields a value that is at most $\mathcal{O}(\max\{\epsilon, e\})$ distance away from the optimal value, and it requires at most $\mathcal{O}(\epsilon^{-3})$ samples to arrive at the intended local policy for any $\epsilon > 0$.

## 1.2 Related Works

**Single Agent RL:** Tabular algorithms such as Q-learning (Watkins and Dayan, 1992), and SARSA (Rummery and Niranjan, 1994) were the first RL algorithms adopted in single agent learning literature. Due to their scalability issues, however, these algorithms could not be deployed for problems with large state-space.

Recently, neural network based Q-iteration (Mnih et al., 2015), and policy-iteration (Mnih et al., 2016) algorithms have gained popularity as a substitute for tabular procedures. However, they are still inadequate for large scale multi-agent problems due to the exponential blow-up of joint state-space.

**Multi-Agent Local Policy:** As discussed before, one way to introduce scalability into multi-agent learning is to restrict the policies to be local. The easiest and perhaps the most popular way to train the local policies is via Independent Q-learning (IQL) procedure which has been widely adapted in many application scenarios. For example, many adaptive traffic signal control algorithms apply some form of IQL (Wei et al., 2019; Chu et al., 2019). Unlike IQL, centralised training with decentralised execution (CTDE) based algorithms train local policies in a centralised manner. One of the simplest CTDE based training procedure is obtained via value decomposition network (VDN) (Sunehag et al., 2018) where the goal is to maximize the sum of local Q-functions of the agents. Later, QMIX (Rashid et al., 2018) was introduced where the target was to maximize a weighted sum of local Q-functions. Other variants of CTDE based training procedures include WQMIX (Rashid et al., 2020), QTRAN (Son et al., 2019) etc. As clarified before, none of these procedures provide any theoretical guarantee. Very recently, there have been some efforts to theoretically characterize the performance of localised policies. However, these studies either assume the reward functions themselves to be local, thereby facilitating no agent interaction in the reward model (Qu et al., 2020), or allow the policies to take state-information from neighbouring agents as an input, thereby expanding the definition of local policies (Lin et al., 2020; Koppel et al., 2021).

**Mean-Field Control:** An alternate way to bring scalability into multi-agent learning is to apply the concept of mean-field control (MFC). Empirically, MFC based algorithms have been applied in many practical scenarios ranging from congestion control (Wang et al., 2020), ride-sharing (Al-Abbasi et al., 2019) and epidemic management (Watkins et al., 2016). Theoretically, it has been recently shown that both in homogeneous (Gu et al., 2021), and heterogeneous (Mondal et al., 2022a) population of agents, and in population with non-uniform interactions (Mondal et al., 2022b), MFC closely approximates the optimal policy. Moreover, various model-free (Angiuli et al., 2022) and model-based (Pasztor et al., 2021) algorithms have been proposed to solve MFC problems. Alongside standard MARL problems, the concept of MFC based solutions has also been applied to other variants of reinforcement learning (RL) problems. For example, (Gast and Gaujal, 2011) considers a system comprising of a single controller and $N$ bodies. The controller is solely responsible for taking actions and the bodies are associated with states that change as functions of the chosen action. At each time instant, the controller receives a reward that solely is a function of the current states. The objective is to strategize the choice of actions as a function of states such that the cumulative reward of the controller is maximized. It is proven that the above problem can be approximated via a mean-field process.

## 2 Model for Cooperative MARL

We consider a collection of $N$ interacting agents. The state of $i$-th agent, $i \in \{1, \cdots, N\}$ at time $t \in \{0, 1, \cdots, \infty\}$ is symbolized as $x_t^i \in \mathcal{X}$ where $\mathcal{X}$ denotes the collection of all possible states (state-space). Each agent also chooses an action at each time instant from a pool of possible actions, $\mathcal{U}$ (action-space). The action of $i$-th agent at time $t$ is indicated as $u_t^i$. The joint state and action of all the agents at time $t$ are denoted as $\boldsymbol{x}_t^N \triangleq \{x_t^i\}_{i=1}^N$ and $\boldsymbol{u}_t^N \triangleq \{u_t^i\}_{i=1}^N$ respectively. The empirical distributions of the joint states, $\boldsymbol{\mu}_t^N$ and actions, $\boldsymbol{\nu}_t^N$ at time $t$ are defined as follows.

$$\boldsymbol{\mu}_t^N(x) = \frac{1}{N} \sum_{i=1}^N \delta(x_t^i = x), \ \forall x \in \mathcal{X} \tag{1}$$

$$\boldsymbol{\nu}_t^N(u) = \frac{1}{N} \sum_{i=1}^N \delta(u_t^i = u), \ \forall u \in \mathcal{U} \tag{2}$$

where $\delta(\cdot)$ denotes the indicator function.

At time $t$, the $i$-th agent receives a reward $r_i(\boldsymbol{x}_t^N, \boldsymbol{u}_t^N)$ and its state changes according to the following state-transition law: $x_{t+1}^i \sim P_i(\boldsymbol{x}_t^N, \boldsymbol{u}_t^N)$. Note that the reward and the transition function not only depend on the

state and action of the associated agent but also on the states, and actions of other agents in the population. This dependence makes the MARL problem difficult to handle. In order to simplify the problem, we assume the reward and transition functions to be of the following form for some $r : \mathcal{X} \times \mathcal{U} \times \mathcal{P}(\mathcal{X}) \times \mathcal{P}(\mathcal{U}) \to \mathbb{R}$, and $P : \mathcal{X} \times \mathcal{U} \times \mathcal{P}(\mathcal{X}) \times \mathcal{P}(\mathcal{U}) \to \mathcal{P}(\mathcal{X})$ where $\mathcal{P}(\cdot)$ is the collection of all probability measures defined over its argument set,

$$r_i(\boldsymbol{x}_t^N, \boldsymbol{u}_t^N) = r(x_t^i, u_t^i, \boldsymbol{\mu}_t^N, \boldsymbol{\nu}_t^N) \tag{3}$$

$$P_i(\boldsymbol{x}_t^N, \boldsymbol{u}_t^N) = P(x_t^i, u_t^i, \boldsymbol{\mu}_t^N, \boldsymbol{\nu}_t^N) \tag{4}$$

$\forall i \in \{1, \cdots, N\}$, $\forall \boldsymbol{x}_t^N \in \mathcal{X}^N$, and $\forall \boldsymbol{u}_t^N \in \mathcal{U}^N$. Two points are worth mentioning. First, (3) and (4) suggest that the reward and transition functions of each agent take the state and action of that agent, along with empirical state and action distributions of the entire population as their arguments. As a result, the agents can influence each other only via the distributions $\boldsymbol{\mu}_t^N$, $\boldsymbol{\nu}_t^N$ which are defined by (1), (2) respectively. Second, the functions $r$, $P$ are taken to be the same for every agent. This makes the population homogeneous. Such assumptions are common in the mean-field literature and holds when the agents are identical and exchangeable.

We define a policy to be a (probabilistic) rule that dictates how a certain agent must choose its action for a given joint state of the population. Mathematically, a policy $\pi$ is a mapping of the following form, $\pi : \mathcal{X}^N \to \mathcal{P}(\mathcal{U})$. Let $\pi_t^i$ denote the policy of $i$-th agent at time $t$. Due to (3) and (4), we can, without loss of generality, write the following equation for some $\pi : \mathcal{X} \times \mathcal{P}(\mathcal{X}) \to \mathcal{U}$.

$$\pi_t^i(\boldsymbol{x}_t^N) = \pi(x_t^i, \boldsymbol{\mu}_t^N) \tag{5}$$

In other words, one can equivalently define a policy to be a rule that dictates how an agent should choose its action given its state and the empirical state-distribution of the entire population. Observe that, due to agent-homogeneity, the policy function, $\pi$, of each agent are the same. With this revised definition, let the policy of any agent at time $t$ be denoted as $\pi_t$, and the sequence of these policies be indicated as $\boldsymbol{\pi} \triangleq \{\pi_t\}_{t \in \{0,1,\cdots\}}$. For a given joint initial state, $\boldsymbol{x}_0^N$, the population-average value of the policy sequence $\boldsymbol{\pi}$ is defined as follows.

$$v_{\text{MARL}}(\boldsymbol{x}_0^N, \boldsymbol{\pi}) \triangleq \frac{1}{N} \sum_{i=1}^{N} \mathbb{E} \left[ \sum_{t=0}^{\infty} \gamma^t r_i(\boldsymbol{x}_t^N, \boldsymbol{u}_t^N) \right] = \frac{1}{N} \sum_{i=1}^{N} \mathbb{E} \left[ \sum_{t=0}^{\infty} \gamma^t r(x_t^i, u_t^i, \boldsymbol{\mu}_t^N, \boldsymbol{\nu}_t^N) \right] \tag{6}$$

where the expectation is computed over all state-action trajectories induced by the policy-sequence, $\boldsymbol{\pi}$ and $\gamma \in [0, 1)$ is the discount factor. The target of MARL is to maximize $v_{\text{MARL}}(\boldsymbol{x}_0^N, \cdot)$ over all policy sequences, $\boldsymbol{\pi}$. Let the optimal policy sequence be $\boldsymbol{\pi}_{\text{MARL}}^* \triangleq \{\pi_{t,\text{MARL}}^*\}_{t \in \{0,1,\cdots\}}$. Note that, in order to execute the policy $\pi_{t,\text{MARL}}^*$, in general, the agents must have knowledge about their own states at time $t$ as well as the empirical state-distribution of the entire population at the same instant. As stated previously, the collection of population-wide state information is a costly process in many practical scenarios. In the subsequent sections of this article, our target, therefore, is to identify (a sequence of) local policies that the agents can execute solely with the knowledge of their own states. Additionally, the policy must be such that its value (expected time-discounted cumulative reward) is close to that generated by the optimal policy-sequence, $\boldsymbol{\pi}_{\text{MARL}}^*$.

## 3 The Mean-Field Control (MFC) Framework

Mean-Field Control (MFC) considers a scenario where the agent population size is infinite. Due to the homogeneity of the agents, in such a framework, it is sufficient to track only one representative agent. We denote the state and action of the representative agent at time $t$ as $x_t \in \mathcal{X}$ and $u_t \in \mathcal{U}$, respectively, while the state and action distributions of the infinite population at the same instance are indicated as $\boldsymbol{\mu}_t^\infty \in \mathcal{P}(\mathcal{X})$ and $\boldsymbol{\nu}_t^\infty \in \mathcal{P}(\mathcal{U})$, respectively. For a given sequence of policies $\boldsymbol{\pi} \triangleq \{\pi_t\}_{t \in \{0,1,\cdots\}}$ and the state distribution $\boldsymbol{\mu}_t^\infty$ at time $t$, the action distribution $\boldsymbol{\nu}_t^\infty$ at the same instant can be computed as follows.

$$\boldsymbol{\nu}_t^\infty = \nu^{\text{MF}}(\boldsymbol{\mu}_t^\infty, \pi_t) \triangleq \sum_{x \in \mathcal{X}} \pi_t(x, \boldsymbol{\mu}_t^\infty) \boldsymbol{\mu}_t^\infty(x) \tag{7}$$

In a similar fashion, the state-distribution at time $t + 1$ can be evaluated as shown below.

$$\boldsymbol{\mu}_{t+1}^{\infty} = P^{\mathrm{MF}}(\boldsymbol{\mu}_t^{\infty}, \pi_t) \triangleq \sum_{x \in \mathcal{X}} \sum_{u \in \mathcal{U}} P(x, u, \boldsymbol{\mu}_t^{\infty}, \nu^{\mathrm{MF}}(\boldsymbol{\mu}_t^{\infty}, \pi_t)) \pi_t(x, \boldsymbol{\mu}_t^{\infty})(u) \boldsymbol{\mu}_t^{\infty}(x) \tag{8}$$

Finally, the average reward at time $t$ can be expressed as follows.

$$r^{\mathrm{MF}}(\boldsymbol{\mu}_t^{\infty}, \pi_t) \triangleq \sum_{x \in \mathcal{X}} \sum_{u \in \mathcal{U}} r(x, u, \boldsymbol{\mu}_t^{\infty}, \nu^{\mathrm{MF}}(\boldsymbol{\mu}_t^{\infty}, \pi_t)) \pi_t(x, \boldsymbol{\mu}_t^{\infty})(u) \boldsymbol{\mu}_t^{\infty}(x) \tag{9}$$

For an initial state distribution, $\boldsymbol{\mu}_0$, the value of a policy sequence $\boldsymbol{\pi} = \{\pi_t\}_{t \in \{0,1,\cdots\}}$ is computed as shown below.

$$v_{\mathrm{MF}}(\boldsymbol{\mu}_0, \boldsymbol{\pi}) = \sum_{t=0}^{\infty} \gamma^t r^{\mathrm{MF}}(\boldsymbol{\mu}_t^{\infty}, \pi_t) \tag{10}$$

The goal of Mean-Field Control is to maximize $v_{\mathrm{MF}}(\boldsymbol{\mu}_0, \cdot)$ over all policy sequences $\boldsymbol{\pi}$. Let the optimal policy sequence be denoted as $\boldsymbol{\pi}_{\mathrm{MF}}^*$. (Gu et al., 2021) recently established that if the agents execute $\boldsymbol{\pi}_{\mathrm{MF}}^*$ in the $N$-agent MARL system, then the $N$-agent value function generated by this policy-sequence well-approximates the value function generated by $\boldsymbol{\pi}_{\mathrm{MARL}}^*$ when $N$ is large. However, to execute the sequence $\boldsymbol{\pi}_{\mathrm{MF}}^*$ in an $N$-agent system, each agent must be aware of empirical state distributions, $\{\boldsymbol{\mu}_t^N\}_{t \in \{0,1,\cdots\}}$, along with its own states at each time instant. In other words, when the $i$-th agent in an $N$-agent system executes the policy sequence $\boldsymbol{\pi}_{\mathrm{MF}}^* \triangleq \{\pi_{t,\mathrm{MF}}^*\}_{t \in \{0,1,\cdots\}}$, it chooses action at time $t$ according to the following distribution, $u_t^i \sim \pi_{t,\mathrm{MF}}^*(x_t^i, \boldsymbol{\mu}_t^N)$. As stated in Section 2, the computation of empirical state distribution of the whole population at each instant is a costly procedure. In the following subsection, we discuss how we can design a near-optimal policy that does not require $\{\boldsymbol{\mu}_t^N\}_{t \in \{1,2,\cdots\}}$ its execution.

**Designing Local Policies:** Note from Eq. (8) that the evolution of infinite-population state distribution, $\boldsymbol{\mu}_t^{\infty}$ is a deterministic equation. Hence, if the initial distribution $\boldsymbol{\mu}_0$ is disseminated among the agents, then each agent can locally compute the distributions, $\{\boldsymbol{\mu}_t^{\infty}\}$ for $t > 0$. Consider the following policy sequence, denoted as $\tilde{\boldsymbol{\pi}}_{\mathrm{MF}}^* \triangleq \{\tilde{\pi}_{t,\mathrm{MF}}^*\}_{t \in \{0,1,\cdots\}}$.

$$\tilde{\pi}_{t,\mathrm{MF}}^*(x, \boldsymbol{\mu}) \triangleq \pi_{t,\mathrm{MF}}^*(x, \boldsymbol{\mu}_t^{\infty}), \ \forall x \in \mathcal{X}, \forall \boldsymbol{\mu} \in \mathcal{P}(\mathcal{X}), \forall t \in \{0, 1, \cdots\} \tag{11}$$

The sequence, $\boldsymbol{\pi}_{\mathrm{MF}}^* \triangleq \{\pi_{t,\mathrm{MF}}^*\}_{t \in \{0,1,\cdots\}}$, as mentioned before, is the optimal policy-sequence that maximizes the mean-field value function, $v_{\mathrm{MF}}(\boldsymbol{\mu}_0^{\infty}, \cdot)$ for a given initial state distribution, $\boldsymbol{\mu}_0$. Note that, in order to execute the policy-sequence, $\tilde{\pi}_{t,\mathrm{MF}}^*$ in an $N$-agent system, each agent must be aware of its own state as well as the state distribution of an infinite agent system at each time instant—both of which can be obtained locally provided it is aware of the initial distribution, $\boldsymbol{\mu}_0$. In other words, the policy-sequence $\tilde{\boldsymbol{\pi}}_{\mathrm{MF}}^*$ completely disregards the empirical distributions, $\{\boldsymbol{\mu}_t^N\}_{t \in \{1,2,\cdots\}}$ (instead depends on $\{\boldsymbol{\mu}_t^{\infty}\}_{t \in \{1,2,\cdots\}}$), and therefore is locally executable for $t > 0$.

The above discussion well establishes $\tilde{\boldsymbol{\pi}}_{\mathrm{MF}}^*$ as a locally executable policy. However, it is not clear at this point how well $\tilde{\boldsymbol{\pi}}_{\mathrm{MF}}^*$ performs in comparison to the optimal policy, $\boldsymbol{\pi}_{\mathrm{MARL}}^*$ in an $N$-agent system. Must one embrace a significant performance deterioration to enforce locality? In the next section, we provide an answer to this crucial question.

# 4 Main Result

Before stating the main result, we shall state a few assumptions that are needed to establish it. Our first assumption is on the reward function, $r$, and the state-transition function, $P$, defined in (3) and (4), respectively.

**Assumption 1.** *The reward function, $r$, is bounded and Lipschitz continuous with respect to the mean-field arguments. Mathematically, there exists $M_R, L_R > 0$ such that the following holds*

*(a)* $|r(x, u, \boldsymbol{\mu}, \boldsymbol{\nu})| \leq M_R$
*(b)* $|r(x, u, \boldsymbol{\mu}_1, \boldsymbol{\nu}_1) - r(x, u, \boldsymbol{\mu}_2, \boldsymbol{\nu}_2)| \leq L_R [|\boldsymbol{\mu}_1 - \boldsymbol{\mu}_2|_1 + |\boldsymbol{\nu}_1 - \boldsymbol{\nu}_2|_1]$

$\forall x \in \mathcal{X}, \forall u \in \mathcal{U}, \forall \boldsymbol{\mu}_1, \boldsymbol{\mu}_2 \in \mathcal{P}(\mathcal{X})$, *and* $\forall \boldsymbol{\nu}_1, \boldsymbol{\nu}_2 \in \mathcal{P}(\mathcal{U})$. *The function* $|\cdot|$ *denotes* $L_1$-*norm.*

**Assumption 2.** *The state-transition function, $P$, is Lipschitz continuous with respect to the mean-field arguments. Mathematically, there exists $L_P > 0$ such that the following holds*

*(a)* $|P(x, u, \boldsymbol{\mu}_1, \boldsymbol{\nu}_1) - P(x, u, \boldsymbol{\mu}_2, \boldsymbol{\nu}_2)| \leq L_P [|\boldsymbol{\mu}_1 - \boldsymbol{\mu}_2|_1 + |\boldsymbol{\nu}_1 - \boldsymbol{\nu}_2|_1]$

$\forall x \in \mathcal{X}, \forall u \in \mathcal{U}, \forall \boldsymbol{\mu}_1, \boldsymbol{\mu}_2 \in \mathcal{P}(\mathcal{X})$, *and* $\forall \boldsymbol{\nu}_1, \boldsymbol{\nu}_2 \in \mathcal{P}(\mathcal{U})$.

Assumptions 1 and 2 ensure the boundedness and the Lipschitz continuity of the reward and state transition functions. The very definition of transition function makes it bounded. So, it is not explicitly mentioned in Assumption 2. These assumptions are commonly taken in the mean-field literature (Mondal et al., 2022a; Gu et al., 2021). Our next assumption is on the class of policy functions.

**Assumption 3.** *The class of admittable policy functions, $\Pi$, is such that every $\pi \in \Pi$ satisfies the following for some $L_Q > 0$,*

$$|\pi(x, \boldsymbol{\mu}_1) - \pi(x, \boldsymbol{\mu}_2)|_1 \leq L_Q |\boldsymbol{\mu}_1 - \boldsymbol{\mu}_2|_1$$

$\forall x \in \mathcal{X}, \forall \boldsymbol{\mu}_1, \boldsymbol{\mu}_2 \in \mathcal{P}(\mathcal{X})$.

Assumption 3 states that every policy is presumed to be Lipschitz continuous w. r. t. the mean-field state distribution argument. This assumption is also common in the literature (Carmona et al., 2018; Pasztor et al., 2021) and typically satisfied by Neural Network based policies with bounded weights. Corresponding to the admittable policy class $\Pi$, we define $\Pi_\infty \triangleq \Pi \times \Pi \times \cdots$ as the set of all admittable policy-sequences.

We now state our main result. The proof of Theorem 1 is relegated to Appendix A.

**Theorem 1.** *Let $\boldsymbol{x}_0^N$ be the initial states in an $N$-agent system and $\boldsymbol{\mu}_0$ be its associated empirical distribution. Assume $\boldsymbol{\pi}_{\mathrm{MF}}^* \triangleq \{\pi_{t,\mathrm{MF}}^*\}_{t \in \{0,1,\cdots\}}$, and $\boldsymbol{\pi}_{\mathrm{MARL}}^* \triangleq \{\pi_{t,\mathrm{MARL}}^*\}_{t \in \{0,1,\cdots\}}$ to be policy sequences that maximize $v_{\mathrm{MF}}(\boldsymbol{\mu}_0, \cdot)$, and $v_{\mathrm{MARL}}(\boldsymbol{x}_0^N, \cdot)$ respectively within the class, $\Pi_\infty$. Let $\tilde{\boldsymbol{\pi}}_{\mathrm{MF}}^*$ be a localized policy sequence corresponding to $\boldsymbol{\pi}_{\mathrm{MF}}^*$ which is defined by (11). If Assumptions 1 and 2 hold, then the following is true whenever $\gamma S_P < 1$.*

$$\begin{aligned}
|v_{\mathrm{MARL}}(\boldsymbol{x}_0^N, \boldsymbol{\pi}_{\mathrm{MARL}}^*) - v_{\mathrm{MARL}}(\boldsymbol{x}_0^N, \tilde{\boldsymbol{\pi}}_{\mathrm{MF}}^*)| &\leq \left(\frac{2}{1-\gamma}\right) \left[\frac{M_R}{\sqrt{N}} + \frac{L_R}{\sqrt{N}} \sqrt{|\mathcal{U}|}\right] \\
&+ \frac{1}{\sqrt{N}} \left[\sqrt{|\mathcal{X}|} + \sqrt{|\mathcal{U}|}\right] \left(\frac{2 S_R C_P}{S_P - 1}\right) \left[\frac{1}{1 - \gamma S_P} - \frac{1}{1-\gamma}\right]
\end{aligned} \tag{12}$$

*The parameters are defined as follows* $C_P \triangleq 2 + L_P$, $S_R \triangleq (M_R + 2L_R) + L_Q(M_R + L_R)$, *and* $S_P \triangleq (1 + 2L_P) + L_Q(1 + L_P)$ *where* $M_R, L_R, L_P$, *and* $L_Q$ *are defined in Assumptions* $1 - 3$.

Theorem 1 accomplishes our goal of showing that, for large $N$, there exists a locally executable policy whose $N$-agent value function is at most $\mathcal{O}\left(\frac{1}{\sqrt{N}} \left[\sqrt{|\mathcal{X}|} + \sqrt{|\mathcal{U}|}\right]\right)$ distance away from the optimal $N$-agent value function. Note that the optimality gap decreases with increase in $N$. We, therefore, conclude that for large population, locally-executable policies are near-optimal. Interestingly, this result also shows that the optimality gap increases with $|\mathcal{X}|, |\mathcal{U}|$, the sizes of state, and action spaces. In the next section, we prove that, if the reward, and transition functions do not depend on the action distribution of the population, then it is possible to further improve the optimality gap.

# 5 Optimality Error Improvement in a Special Case

The key to improving the optimality gap of Theorem 1 hinges on the following assumption.

**Assumption 4.** *The reward function, $r$, and transition function, $P$ are independent of the action distribution of the population. Mathematically,*

$$(a)\ r(x, u, \boldsymbol{\mu}, \boldsymbol{\nu}) = r(x, u, \boldsymbol{\mu})$$
$$(b)\ P(x, u, \boldsymbol{\mu}, \boldsymbol{\nu}) = P(x, u, \boldsymbol{\mu})$$

$\forall x \in \mathcal{X}$, $\forall u \in \mathcal{U}$, $\forall \boldsymbol{\mu} \in \mathcal{P}(\mathcal{X})$, and $\forall \boldsymbol{\nu} \in \mathcal{P}(\mathcal{U})$.

Assumption 4 considers a scenario where the reward function, $r$, and the transition function, $P$ is independent of the action distribution. In such a case, the agents can influence each other only through the state distribution. Clearly, this is a special case of the general model considered in Section 2. We would like to clarify that although $r$ and $P$ are assumed to be independent of the action distribution of the population, they still might depend on the action taken by the individual agents. Assumption 4 is commonly assumed in many mean-field related articles (Tiwari et al., 2019).

Theorem 2 (stated below) formally dictates the optimality gap achieved by the local policies under Assumption 4. The proof of Theorem 2 is relegated to Appendix B.

**Theorem 2.** *Let $\boldsymbol{x}_0^N$ be the initial states in an $N$-agent system and $\boldsymbol{\mu}_0$ be its associated empirical distribution. Assume $\boldsymbol{\pi}_{\mathrm{MF}}^* \triangleq \{\pi_{t,\mathrm{MF}}^*\}_{t\in\{0,1,\cdots\}}$, and $\boldsymbol{\pi}_{\mathrm{MARL}}^* \triangleq \{\pi_{t,\mathrm{MARL}}^*\}_{t\in\{0,1,\cdots\}}$ to be policy sequences that maximize $v_{\mathrm{MF}}(\boldsymbol{\mu}_0, \cdot)$, and $v_{\mathrm{MARL}}(\boldsymbol{x}_0^N, \cdot)$ respectively within the class, $\Pi_\infty$. Let $\tilde{\boldsymbol{\pi}}_{\mathrm{MF}}^*$ be a localized policy sequence corresponding to $\boldsymbol{\pi}_{\mathrm{MF}}^*$ which is defined by (11). If Assumptions 1, 2, and 4 hold, then the following is true whenever $\gamma S_P < 1$.*

$$|v_{\mathrm{MARL}}(\boldsymbol{x}_0^N, \boldsymbol{\pi}_{\mathrm{MARL}}^*) - v_{\mathrm{MARL}}(\boldsymbol{x}_0^N, \tilde{\boldsymbol{\pi}}_{\mathrm{MF}}^*)| \leq \left(\frac{2}{1-\gamma}\right)\left[\frac{M_R}{\sqrt{N}}\right]$$
$$+ \frac{1}{\sqrt{N}}\sqrt{|\mathcal{X}|}\left(\frac{4S_R}{S_P - 1}\right)\left[\frac{1}{1 - \gamma S_P} - \frac{1}{1-\gamma}\right] \tag{13}$$

*The parameters are same as in Theorem 1.*

Theorem 2 suggests that, under Assumption 4, the optimality gap for local policies can be bounded as $\mathcal{O}(\frac{1}{\sqrt{N}}\sqrt{|\mathcal{X}|})$. Although the dependence of the optimality gap on $N$ is same as in Theorem 1, its dependence on the size of state, action spaces has been reduced to $\mathcal{O}(\sqrt{|\mathcal{X}|})$ from $\mathcal{O}(\sqrt{|\mathcal{X}|} + \sqrt{|\mathcal{U}|})$ stated previously. This result is particularly useful for applications where Assumption 4 holds, and the size of action-space is large.

A natural question that might arise in this context is whether it is possible to remove the dependence of the optimality gap on the size of state-space, $|\mathcal{X}|$ by imposing the restriction that $r$, $P$ are independent of the state-distribution. Despite our effort, we could not arrive at such a result. This points towards an inherent asymmetry between the roles played by state, and action spaces in mean-field approximation.

# 6 Roadmap of the Proof

In this section, we provide an outline of the proof of Theorem 1. The proof of Theorem 2 is similar. The goal in the proof of Theorem 1 is to establish the following three bounds.

$$G_0 \triangleq |v_{\mathrm{MARL}}(\boldsymbol{x}_0^N, \boldsymbol{\pi}_{\mathrm{MARL}}^*) - v_{\mathrm{MF}}(\boldsymbol{\mu}_0, \boldsymbol{\pi}_{\mathrm{MARL}}^*)| = \mathcal{O}\left(\frac{\sqrt{|\mathcal{X}|} + \sqrt{|\mathcal{U}|}}{\sqrt{N}}\right) \tag{14}$$

$$G_1 \triangleq |v_{\mathrm{MARL}}(\boldsymbol{x}_0^N, \boldsymbol{\pi}_{\mathrm{MF}}^*) - v_{\mathrm{MF}}(\boldsymbol{\mu}_0, \boldsymbol{\pi}_{\mathrm{MF}}^*)| = \mathcal{O}\left(\frac{\sqrt{|\mathcal{X}|} + \sqrt{|\mathcal{U}|}}{\sqrt{N}}\right) \tag{15}$$

$$G_2 \triangleq |v_{\mathrm{MARL}}(\boldsymbol{x}_0^N, \tilde{\boldsymbol{\pi}}_{\mathrm{MF}}^*) - v_{\mathrm{MF}}(\boldsymbol{\mu}_0, \boldsymbol{\pi}_{\mathrm{MF}}^*)| = \mathcal{O}\left(\frac{\sqrt{|\mathcal{X}|} + \sqrt{|\mathcal{U}|}}{\sqrt{N}}\right) \tag{16}$$

where $v_{\text{MARL}}(\cdot, \cdot)$ and $v_{\text{MF}}(\cdot, \cdot)$ are value functions defined by (6) and (10), respectively. Once these bounds are established, (13) can be proven easily. For example, observe that,

$$
\begin{aligned}
& v_{\text{MARL}}(\boldsymbol{x}_0^N, \boldsymbol{\pi}_{\text{MARL}}^*) - v_{\text{MARL}}(\boldsymbol{x}_0^N, \tilde{\boldsymbol{\pi}}_{\text{MF}}^*) \\
=\ & v_{\text{MARL}}(\boldsymbol{x}_0^N, \boldsymbol{\pi}_{\text{MARL}}^*) - v_{\text{MF}}(\boldsymbol{\mu}_0, \boldsymbol{\pi}_{\text{MF}}^*) + v_{\text{MF}}(\boldsymbol{\mu}_0, \boldsymbol{\pi}_{\text{MF}}^*) - v_{\text{MARL}}(\boldsymbol{x}_0^N, \tilde{\boldsymbol{\pi}}_{\text{MF}}^*) \\
\overset{(a)}{\leq}\ & v_{\text{MARL}}(\boldsymbol{x}_0^N, \boldsymbol{\pi}_{\text{MARL}}^*) - v_{\text{MF}}(\boldsymbol{\mu}_0, \boldsymbol{\pi}_{\text{MARL}}^*) + G_2 \\
\leq\ & G_0 + G_2 = \mathcal{O}\left( \frac{\sqrt{|\mathcal{X}|} + \sqrt{|\mathcal{U}|}}{\sqrt{N}} \right)
\end{aligned}
\tag{17}
$$

Inequality (a) uses the fact that $\boldsymbol{\pi}_{\text{MF}}^*$ is a maximizer of $v_{\text{MF}}(\boldsymbol{\mu}_0, \cdot)$. Following a similar argument, the term $v_{\text{MARL}}(\boldsymbol{x}_0^N, \tilde{\boldsymbol{\pi}}_{\text{MF}}^*) - v_{\text{MARL}}(\boldsymbol{x}_0^N, \boldsymbol{\pi}_{\text{MARL}}^*)$ can be bounded by $G_1 + G_2 = \mathcal{O}\left( \frac{\sqrt{|\mathcal{X}|} + \sqrt{|\mathcal{U}|}}{\sqrt{N}} \right)$. Combining with (17), we can establish that $|v_{\text{MARL}}(\boldsymbol{x}_0^N, \boldsymbol{\pi}_{\text{MARL}}^*) - v_{\text{MARL}}(\boldsymbol{x}_0^N, \tilde{\boldsymbol{\pi}}_{\text{MF}}^*)| = \mathcal{O}\left( \frac{\sqrt{|\mathcal{X}|} + \sqrt{|\mathcal{U}|}}{\sqrt{N}} \right)$.

To prove (14), (15), and (16), it is sufficient to show that the following holds

$$
J_0 \triangleq |v_{\text{MARL}}(\boldsymbol{x}_0^N, \bar{\boldsymbol{\pi}}) - v_{\text{MF}}(\boldsymbol{\mu}_0, \boldsymbol{\pi})| = \mathcal{O}\left( \frac{\sqrt{|\mathcal{X}|} + \sqrt{|\mathcal{U}|}}{\sqrt{N}} \right)
\tag{18}
$$

for suitable choice of policy-sequences, $\bar{\boldsymbol{\pi}}$, and $\boldsymbol{\pi}$. This is achieved as follows.

- Note that the value functions are defined as the time-discounted sum of average rewards. To bound $J_0$ defined in (18), we therefore focus on the difference between the empirical $N$-agent average reward, $\frac{1}{N} \sum_i r(\bar{x}_t^i, \bar{u}_t^i, \bar{\boldsymbol{\mu}}_t^N, \bar{\boldsymbol{\nu}}_t^N)$ at time $t$ generated from the policy-sequence $\bar{\boldsymbol{\pi}}$, and the infinite agent average reward, $r^{\text{MF}}(\boldsymbol{\mu}_t^\infty, \pi_t)$ at the same instant generated by $\boldsymbol{\pi}$. The parameters $\bar{x}_t^i, \bar{u}_t^i, \bar{\boldsymbol{\mu}}_t^N, \bar{\boldsymbol{\nu}}_t^N$ respectively denote state, action of $i$-th agent, and empirical state, action distributions of $N$-agent system at time $t$ corresponding to the policy-sequence $\bar{\boldsymbol{\pi}}$. Also, by $\bar{\boldsymbol{\mu}}_t^\infty$, $\boldsymbol{\mu}_t^\infty$, we denote the infinite agent state distributions at time $t$ corresponding to $\bar{\boldsymbol{\pi}}$, $\boldsymbol{\pi}$ respectively.

- The difference between $\frac{1}{N} \sum_i r(\bar{x}_t^i, \bar{u}_t^i, \bar{\boldsymbol{\mu}}_t^N, \bar{\boldsymbol{\nu}}_t^N)$ and $r^{\text{MF}}(\boldsymbol{\mu}_t^\infty, \pi_t)$ can be upper bounded by three terms.

- The first term is $|\frac{1}{N} \sum_i r(\bar{x}_t^i, \bar{u}_t^i, \bar{\boldsymbol{\mu}}_t^N, \bar{\boldsymbol{\nu}}_t^N) - r^{\text{MF}}(\bar{\boldsymbol{\mu}}_t^N, \bar{\pi}_t)|$ which is bounded as $\mathcal{O}(\frac{1}{N}\sqrt{|\mathcal{U}|})$ by Lemma 7 (stated in Appendix A.2).

- The second term is $|r^{\text{MF}}(\bar{\boldsymbol{\mu}}_t^N, \bar{\pi}_t) - r^{\text{MF}}(\bar{\boldsymbol{\mu}}_t^\infty, \bar{\pi}_t)|$ which can be bounded as $\mathcal{O}(|\bar{\boldsymbol{\mu}}_t^N - \bar{\boldsymbol{\mu}}_t^\infty|)$ by using the Lipschitz continuity of $r^{\text{MF}}$ established in Lemma 4 in Appendix A.1. The term $|\bar{\boldsymbol{\mu}}_t^N - \bar{\boldsymbol{\mu}}_t^\infty|$ can be further bounded as $\mathcal{O}\left( \frac{1}{\sqrt{N}} \left[ \sqrt{|\mathcal{X}|} + \sqrt{|\mathcal{U}|} \right] \right)$ using Lemma 8 stated in Appendix A.2.

- Finally, the third term is $|r^{\text{MF}}(\bar{\boldsymbol{\mu}}_t^\infty, \bar{\pi}_t) - r^{\text{MF}}(\boldsymbol{\mu}_t^\infty, \pi_t)|$. Clearly, if $\bar{\boldsymbol{\pi}} = \boldsymbol{\pi}$, then this term is zero. Moreover, if $\bar{\boldsymbol{\pi}}$ is localization of $\boldsymbol{\pi}$, defined similarly as in (11), then the term is zero as well. This is due to the fact that the trajectory of state and action distributions generated by $\bar{\boldsymbol{\pi}}, \boldsymbol{\pi}$ are exactly the same in an infinite agent system. This, however, may not be true in an $N$-agent system.

- For the above two cases, i.e., when $\bar{\boldsymbol{\pi}}, \boldsymbol{\pi}$ are the same or one is localization of another, we can bound $|\frac{1}{N} \sum_i r(\bar{x}_t^i, \bar{u}_t^i, \bar{\boldsymbol{\mu}}_t^N, \bar{\boldsymbol{\nu}}_t^N) - r^{\text{MF}}(\boldsymbol{\mu}_t^\infty, \pi_t)|$ as $\mathcal{O}\left( \frac{1}{\sqrt{N}} \left[ \sqrt{|\mathcal{X}|} + \sqrt{|\mathcal{U}|} \right] \right)$ by taking a sum of the above three bounds. The bound obtained is, in general, $t$ dependent. By taking a time discounted sum of these bounds, we can establish (18).

- Finally, (14), (15), (16) are established by injecting the following pairs of policy-sequences in (18), $(\bar{\boldsymbol{\pi}}, \boldsymbol{\pi}) = (\boldsymbol{\pi}_{\text{MARL}}^*, \boldsymbol{\pi}_{\text{MARL}}^*), (\boldsymbol{\pi}_{\text{MF}}^*, \boldsymbol{\pi}_{\text{MF}}^*), (\tilde{\boldsymbol{\pi}}_{\text{MF}}^*, \boldsymbol{\pi}_{\text{MF}}^*)$. Note that $\bar{\boldsymbol{\pi}}, \boldsymbol{\pi}$ are equal for first two pairs whereas in the third case, $\bar{\boldsymbol{\pi}}$ is a localization of $\boldsymbol{\pi}$.

# 7 Algorithm to obtain Near-Optimal Local Policy

In section 3, we discussed how near-optimal local policies can be obtained if the optimal mean-field policy sequence $\boldsymbol{\pi}^*_{\mathrm{MF}}$ is known. In this section, we first describe a natural policy gradient (NPG) based algorithm to approximately obtain $\boldsymbol{\pi}^*_{\mathrm{MF}}$. Later, we also provide an algorithm to describe how the obtained policy-sequence is localised and executed in a decentralised manner.

Recall from section 3 that, in an infinite agent system, it is sufficient to track only one representative agent which, at instant $t$, takes an action $u_t$ on the basis of its observation of its own state, $x_t$, and the state distribution, $\boldsymbol{\mu}_t$ of the population. Hence, the evaluation of $\boldsymbol{\pi}^*_{\mathrm{MF}}$ can be depicted as a Markov Decision Problem with state-space $\mathcal{X} \times \mathcal{P}(\mathcal{X})$, and action space $\mathcal{U}$. One can, therefore, presume $\boldsymbol{\pi}^*_{\mathrm{MF}}$ to be stationary i.e., $\boldsymbol{\pi}^*_{\mathrm{MF}} = \{\pi^*_{\mathrm{MF}}, \pi^*_{\mathrm{MF}}, \cdots\}$ (Puterman, 2014). We would like to point out that the same conclusion may not hold for the localised policy-sequence $\tilde{\boldsymbol{\pi}}^*_{\mathrm{MF}}$. Stationarity of the sequence, $\boldsymbol{\pi}^*_{\mathrm{MF}}$ reduces our task to finding an optimal policy, $\pi^*_{\mathrm{MF}}$. This facilitates a drastic reduction in the search space. With slight abuse of notation, in this section, we shall use $\pi_\Phi$ to denote a policy parameterized by $\Phi$, as well as the stationary sequence generated by it.

Let the collection of policies be denoted as $\Pi$. We shall assume that $\Pi$ is parameterized by $\Phi \in \mathbb{R}^{\mathrm{d}}$. Consider an arbitrary policy $\pi_\Phi \in \Pi$. The Q-function associated with this policy be defined as follows $\forall x \in \mathcal{X}$, $\forall \boldsymbol{\mu} \in \mathcal{P}(\mathcal{X})$, and $\forall u \in \mathcal{U}$.

$$Q_\Phi(x, \boldsymbol{\mu}, u) \triangleq \mathbb{E}\left[\sum_{t=0}^{\infty} \gamma^t r(x_t, u_t, \boldsymbol{\mu}_t, \boldsymbol{\nu}_t) \Big| x_0 = x, \boldsymbol{\mu}_0 = \boldsymbol{\mu}, u_0 = u\right] \tag{19}$$

where $u_{t+1} \sim \pi_\Phi(x_{t+1}, \boldsymbol{\mu}_{t+1})$, $x_{t+1} \sim P(x_t, u_t, \boldsymbol{\mu}_t, \boldsymbol{\nu}_t)$, and $\boldsymbol{\mu}_t, \boldsymbol{\nu}_t$ are recursively obtained $\forall t > 0$ using $(8), (7)$ respectively from the initial distribution $\boldsymbol{\mu}_0 = \boldsymbol{\mu}$. The advantage function associated with $\pi_\Phi$ is defined as shown below.

$$A_\Phi(x, \boldsymbol{\mu}, u) \triangleq Q_\Phi(x, \boldsymbol{\mu}, u) - \mathbb{E}[Q_\Phi(x, \boldsymbol{\mu}, \bar{u})] \tag{20}$$

The expectation is evaluated over $\bar{u} \sim \pi_\Phi(x, \boldsymbol{\mu})$. To obtain the optimal policy, $\pi^*_{\mathrm{MF}}$, we apply the NPG update (Agarwal et al., 2021; Liu et al., 2020) as shown below with learning parameter, $\eta$. This generates a sequence of parameters $\{\Phi_j\}_{j=1}^J$ from an arbitrary initial choice, $\Phi_0$.

$$\Phi_{j+1} = \Phi_j + \eta \mathbf{w}_j, \mathbf{w}_j \triangleq \arg\min_{\mathbf{w} \in \mathbb{R}^{\mathrm{d}}} L_{\zeta^{\Phi_j}_{\boldsymbol{\mu}_0}}(\mathbf{w}, \Phi_j) \tag{21}$$

The term $\zeta^{\Phi_j}_{\boldsymbol{\mu}_0}$ is the occupancy measure defined as,

$$\zeta^{\Phi_j}_{\boldsymbol{\mu}_0}(x, \boldsymbol{\mu}, u) \triangleq \sum_{\tau=0}^{\infty} \gamma^\tau \mathbb{P}(x_\tau = x, \boldsymbol{\mu}_\tau = \boldsymbol{\mu}, u_\tau = u | x_0 = x, \boldsymbol{\mu}_0 = \boldsymbol{\mu}, u_0 = u, \pi_{\Phi_j})(1 - \gamma)$$

whereas the function $L_{\zeta^{\Phi_j}_{\boldsymbol{\mu}_0}}$ is given as follows.

$$L_{\zeta^{\Phi_j}_{\boldsymbol{\mu}_0}}(\mathbf{w}, \Phi) \triangleq \mathbb{E}_{(x, \boldsymbol{\mu}, u) \sim \zeta^{\Phi_j}_{\boldsymbol{\mu}_0}}\left[\left(A_\Phi(x, \boldsymbol{\mu}, u) - (1 - \gamma)\mathbf{w}^{\mathrm{T}}\nabla_\Phi \log \pi_\Phi(x, \boldsymbol{\mu})(u)\right)^2\right]$$

Note that in $j$-th iteration in (21), the gradient direction $\mathbf{w}_j$ is computed by solving another minimization problem. We employ a stochastic gradient descent (SGD) algorithm to solve this sub-problem. The update equation for the SGD is as follows: $\mathbf{w}_{j,l+1} = \mathbf{w}_{j,l} - \alpha \mathbf{h}_{j,l}$ (Liu et al., 2020) where $\alpha$ is the learning parameter and the gradient direction, $\mathbf{h}_{j,l}$ is defined as follows.

$$\mathbf{h}_{j,l} \triangleq \left(\mathbf{w}^{\mathrm{T}}_{j,l}\nabla_{\Phi_j} \log \pi_{\Phi_j}(x, \boldsymbol{\mu})(u) - \frac{1}{1 - \gamma}\hat{A}_{\Phi_j}(x, \boldsymbol{\mu}, u)\right)\nabla_{\Phi_j} \log \pi_{\Phi_j}(x, \boldsymbol{\mu})(u) \tag{22}$$

---

**Algorithm 1** Natural Policy Gradient Algorithm to obtain the Optimal Policy

---

**Input:** $\eta, \alpha$: Learning rates, $J, L$: Number of execution steps
$\mathbf{w}_0, \Phi_0$: Initial parameters, $\boldsymbol{\mu}_0$: Initial state distribution
**Initialization:** $\Phi \leftarrow \Phi_0$

1: **for** $j \in \{0, 1, \cdots, J-1\}$ **do**
2:     $\mathbf{w}_{j,0} \leftarrow \mathbf{w}_0$
3:     **for** $l \in \{0, 1, \cdots, L-1\}$ **do**
4:         Sample $(x, \boldsymbol{\mu}, u) \sim \zeta_{\boldsymbol{\mu}_0}^{\Phi_j}$ and $\hat{A}_{\Phi_j}(x, \boldsymbol{\mu}, u)$ using Algorithm 3
5:         Compute $\mathbf{h}_{j,l}$ using (22)
        $\mathbf{w}_{j,l+1} \leftarrow \mathbf{w}_{j,l} - \alpha \mathbf{h}_{j,l}$
6:     **end for**
7:     $\mathbf{w}_j \leftarrow \dfrac{1}{L} \sum_{l=1}^{L} \mathbf{w}_{j,l}$
8:     $\Phi_{j+1} \leftarrow \Phi_j + \eta \mathbf{w}_j$
9: **end for**

**Output:** $\{\Phi_1, \cdots, \Phi_J\}$: Policy parameters

---

where $(x, \boldsymbol{\mu}, u)$ is sampled from the occupancy measure $\zeta_{\boldsymbol{\mu}_0}^{\Phi_j}$, and $\hat{A}_{\Phi_j}$ is a unbiased estimator of $A_{\Phi_j}$. We detail the process of obtaining the samples and the estimator in Algorithm 3 in the Appendix M. We would like to clarify that Algorithm 3 of (Agarwal et al., 2021) is the foundation for Algorithm 3. The NPG process is summarised in Algorithm 1.

In Algorithm 2, we describe how the policy obtained from Algorithm 1 can be localised and executed by the agents in a decentralised manner. This essentially follows the ideas discussed in section 3. Note that, in order to execute line 3 in Algorithm 2, the agents must be aware of the transition function, $P$. However, the knowledge of the reward function, $r$, is not required.

---

**Algorithm 2** Decentralised Execution of the Policy generated from Algorithm 1

---

**Input:** $\Phi$: Policy parameter from Algorithm 1, $T$: Number of Execution Steps
$\boldsymbol{\mu}_0$: Initial state distribution, $x_0$: Initial state of the agent.

1: **for** $t \in \{0, 1, \cdots, T-1\}$ **do**
2:     Execute $u_t \sim \pi_\Phi(x_t, \boldsymbol{\mu}_t)$
3:     Compute $\boldsymbol{\mu}_{t+1}$ via mean-field dynamics (8).
4:     Observe the next state $x_{t+1}$ (Updated via $N$-agent dynamics: $x_{t+1} \sim P(x_t, u_t, \boldsymbol{\mu}_t^N, \boldsymbol{\nu}_t^N)$)
5:     Update: $\boldsymbol{\mu}_t \leftarrow \boldsymbol{\mu}_{t+1}$
6:     Update: $x_t \leftarrow x_{t+1}$
7: **end for**

---

Theorem 1 showed that the localization of the optimal mean-field policy $\pi_{\mathrm{MF}}^*$ is near-optimal for large $N$. However, Algorithm 1 can provide only an approximation of $\pi_{\mathrm{MF}}^*$. One might naturally ask: is the decentralised version of the policy given by Algorithm 1 still near-optimal? We provide an answer to this question in Theorem 3. Lemma 1, which follows from Theorem 4.9 of (Liu et al., 2020), is an essential ingredient of this result. The proof of Lemma 1, however, hinges on the assumptions stated below. These are similar to Assumptions 2.1, 4.2, and 4.4 respectively in (Liu et al., 2020).

**Assumption 5.** $\forall \Phi \in \mathbb{R}^d$, $\forall \boldsymbol{\mu}_0 \in \mathcal{P}(\mathcal{X})$, for some $\chi > 0$, $F_{\boldsymbol{\mu}_0}(\Phi) - \chi I_d$ is positive semi-definite where $F_{\boldsymbol{\mu}_0}(\Phi)$ is given as follows.

$$F_{\boldsymbol{\mu}_0}(\Phi) \triangleq \mathbb{E}_{(x, \boldsymbol{\mu}, u) \sim \zeta_{\boldsymbol{\mu}_0}^{\Phi}} \left[ \{\nabla_\Phi \pi_\Phi(x, \boldsymbol{\mu})(u)\} \times \{\nabla_\Phi \log \pi_\Phi(x, \boldsymbol{\mu})(u)\}^{\mathrm{T}} \right]$$

**Assumption 6.** $\forall \Phi \in \mathbb{R}^d$, $\forall \boldsymbol{\mu} \in \mathcal{P}(\mathcal{X})$, $\forall x \in \mathcal{X}$, $\forall u \in \mathcal{U}$,

$$|\nabla_\Phi \log \pi_\Phi(x, \boldsymbol{\mu})(u)|_1 \leq G$$

for some positive constant $G$.

**Assumption 7.** $\forall \Phi_1, \Phi_2 \in \mathbb{R}^d$, $\forall \boldsymbol{\mu} \in \mathcal{P}(\mathcal{X})$, $\forall x \in \mathcal{X}$, $\forall u \in \mathcal{U}$,

$$|\nabla_{\Phi_1} \log \pi_{\Phi_1}(x, \boldsymbol{\mu})(u) - \nabla_{\Phi_2} \log \pi_{\Phi_2}(x, \boldsymbol{\mu})(u)|_1 \leq M|\Phi_1 - \Phi_2|_1$$

*for some positive constant $M$.*

**Assumption 8.** $\forall \Phi \in \mathbb{R}^d$, $\forall \boldsymbol{\mu}_0 \in \mathcal{P}(\mathcal{X})$,

$$L_{\zeta_{\boldsymbol{\mu}_0}^{\Phi^*}}(\mathbf{w}_\Phi^*, \Phi) \leq \epsilon_{\text{bias}}, \quad \mathbf{w}_\Phi^* \triangleq \arg\min_{\mathbf{w} \in \mathbb{R}^d} L_{\zeta_{\boldsymbol{\mu}_0}^{\Phi}}(\mathbf{w}, \Phi)$$

*where $\Phi^*$ is the parameter of the optimal policy.*

The parameter $\epsilon_{\text{bias}}$ indicates the expressive power of the parameterized policy class, $\Pi$. For example, $\epsilon_{\text{bias}} = 0$ for softmax policies, and is small for rich neural network based policies.

**Lemma 1.** *Assume that $\{\Phi_j\}_{j=1}^J$ is the sequence of policy parameters generated from Algorithm 1. If Assumptions $5-8$ hold, then the following inequality holds for some choice of $\eta, \alpha, J, L$, for arbitrary initial parameter $\Phi_0$ and initial state distribution $\boldsymbol{\mu}_0 \in \mathcal{P}(\mathcal{X})$.*

$$\sup_{\Phi \in \mathbb{R}^d} v_{\text{MF}}(\boldsymbol{\mu}_0, \pi_\Phi) - \frac{1}{J}\sum_{j=1}^J v_{\text{MF}}(\boldsymbol{\mu}_0, \pi_{\Phi_j}) \leq \frac{\sqrt{\epsilon_{\text{bias}}}}{1-\gamma} + \epsilon, \tag{23}$$

*The sample complexity of Algorithm 1 to achieve (23) is $\mathcal{O}(\epsilon^{-3})$.*

We now state the following result.

**Theorem 3.** *Let $\boldsymbol{x}_0^N$ be the initial states in an $N$-agent system and $\boldsymbol{\mu}_0$ be its associated empirical distribution. Assume that $\{\Phi_j\}_{j=1}^J$ are the sequences of policy parameters obtained from Algorithm 1, and the set of policies, $\Pi$ obeys Assumption 3. If Assumptions $1, 2, 5-8$ hold, then for arbitrary $\epsilon > 0$, the following relation holds for certain choices of $\eta, \alpha, J, L$, and arbitrary initial distribution, $\boldsymbol{\mu}_0 \in \mathcal{P}(\mathcal{X})$, and initial parameter, $\Phi_0$,*

$$\left| \sup_{\Phi \in \mathbb{R}^d} v_{\text{MARL}}(\boldsymbol{\mu}_0, \pi_\Phi) - \frac{1}{J}\sum_{j=1}^J v_{\text{MARL}}(\boldsymbol{\mu}_0, \tilde{\pi}_{\Phi_j}) \right| \leq \frac{\sqrt{\epsilon_{\text{bias}}}}{1-\gamma} + C\max\{\epsilon, e\},$$

$$e \triangleq \frac{1}{\sqrt{N}}\left[\sqrt{|\mathcal{X}|} + \sqrt{|\mathcal{U}|}\right] \tag{24}$$

*whenever $\gamma S_P < 1$ where $S_P$ is given in Theorem 1. The term $\tilde{\pi}_{\Phi_j}$ denotes the localization of the policy $\pi_{\Phi_j}$ defined similarly as in (11), and $C$ is a constant. The sample complexity of the process is $\mathcal{O}(\epsilon^{-3})$. Additionally, if Assumption 4 is satisfied, then $e$ in (24) can be reduced to $e = \sqrt{|\mathcal{X}|}/\sqrt{N}$.*

The proof of Theorem 3 is relegated to Appendix N. It states that for any $\epsilon > 0$, Algorithm 1 can generate a policy such that if it is localised and executed in a decentralised manner in an $N$-agent system, then the value generated from this localised policy is at most $\mathcal{O}(\max\{\epsilon, e\})$ distance away from the optimal value function. The sample complexity to obtain such a policy is $\mathcal{O}(\epsilon^{-3})$.

## 8 Experiments

The setup considered for the numerical experiment is taken from (Subramanian and Mahajan, 2019) with slight modifications. We consider a network of $N$ collaborative firms that yield the same product but with varying quality. The product quality of $i$-th firm, $i \in \{1, \cdots, N\}$ at time $t \in \{0, 1, \cdots\}$ is denoted as $x_t^i$ that can take values from the set $\mathcal{Q} \triangleq \{0, \cdots, Q-1\}$. At each instant, each firm has two choices to make. Either it can remain unresponsive (which we denote as action 0) or can invest some money to improve the quality of its product (indicated as action 1). If the action taken by $i$-th firm at time $t$ is denoted as $u_t^i$, then its state-transition law is described by the following equation.

$$x_{t+1}^i = \begin{cases} x_t^i & \text{if } u_t^i = 0 \\ x_t^i + \left\lfloor \chi\left(Q - 1 - x_t^i\right)\left(1 - \frac{\bar{\boldsymbol{\mu}}_t^N}{Q}\right)\right\rfloor & \text{elsewhere} \end{cases} \tag{25}$$

where $\chi$ is a uniform random variable in $[0,1]$, and $\bar{\boldsymbol{\mu}}_t^N$ is the mean of the empirical state distribution, $\boldsymbol{\mu}_t^N$. The intuition behind this transition law can be stated as follows. If the firm remain unresponsive i.e., $u_t^i = 0$, the product quality does not improve and the state remains the same. In contrast, if the firm invests some money, the quality improves probabilistically. However, if the average quality, $\bar{\boldsymbol{\mu}}_t^N$ in the market is high, a significant improvement is difficult to achieve. The factor $(1 - \frac{\bar{\boldsymbol{\mu}}_t^N}{Q})$ describes the resistance to improvement due to higher average quality. The reward function of the $i$-th firm is defined as shown below.

$$r(x_t^i, u_t^i, \boldsymbol{\mu}_t^N, \boldsymbol{\nu}_t^N) = \alpha_R x_t^i - \beta_R \bar{\boldsymbol{\mu}}_t^N - \lambda_R u_t^i \tag{26}$$

The first term, $\alpha_R x_t^i$ is due to the revenue earned by the firm; the second term, $\beta_R \bar{\boldsymbol{\mu}}_t^N$ is attributed to the resistance to improvement imparted by high average quality; the third term, $\lambda_R u_t^i$ is due to the cost incurred for the investment. Following our previous convention, let $\boldsymbol{\pi}_{\mathrm{MF}}^*$, and $\tilde{\boldsymbol{\pi}}_{\mathrm{MF}}^*$ denote the optimal mean-field policy-sequence and its corresponding local policy-sequence respectively. Using these notations, we define the error for a given joint initial state, $\boldsymbol{x}_0^N$ as follows.

$$\text{error} \triangleq \left| v_{\mathrm{MARL}}(\boldsymbol{x}_0^N, \boldsymbol{\pi}_{\mathrm{MF}}^*) - v_{\mathrm{MARL}}(\boldsymbol{x}_0^N, \tilde{\boldsymbol{\pi}}_{\mathrm{MF}}^*) \right| \tag{27}$$

Fig. 1 plots error as a function of $N$ and $Q$. Evidently, error decreases with $N$ and increases with $Q$. We would like to point out that $\boldsymbol{\pi}_{\mathrm{MF}}^*$, in general, is not a maximizer of $v_{\mathrm{MARL}}(\boldsymbol{x}_0^N, \cdot)$. However, for the reasons stated in Section 1, it is difficult to evaluate the $N$-agent optimal policy-sequence, $\boldsymbol{\pi}_{\mathrm{MARL}}^*$, especially for large $N$. On the other hand, $\boldsymbol{\pi}_{\mathrm{MF}}^*$ can be easily computed via Algorithm 1 and can act as a good proxy for $\boldsymbol{\pi}_{\mathrm{MARL}}^*$. Fig. 1 therefore essentially describes how well the local policies approximate the optimal value function obtained over all policy-sequences in an $N$-agent system.

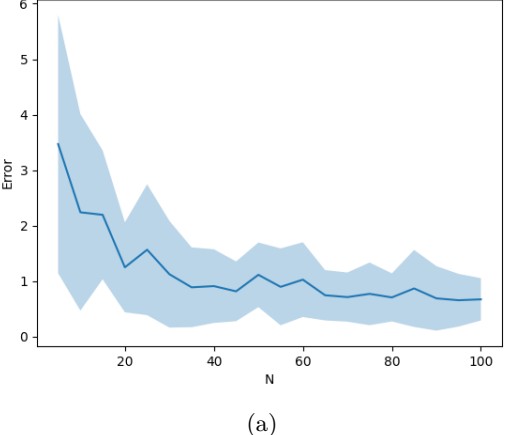

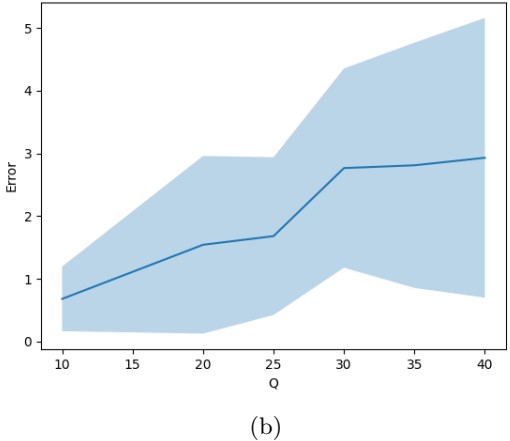

(a)                   (b)

Figure 1: Error defined by (27) as a function of the population size, $N$ (Fig. 1a) and the size of individual state space, $Q$ (Fig. 1b). The solid line, and the half-width of the shaded region respectively denote mean, and standard deviation of error obtained over 25 random seeds. The values of different system parameters are: $\alpha_R = 1$, $\beta_R = \lambda_R = 0.5$. We use $Q = 10$ for Fig. 1a whereas $N = 50$ for Fig. 1b. Moreover, the values of different hyperparameters used in Algorithm 1 are as follows: $\alpha = \eta = 10^{-3}$, $J = L = 10^2$. We use a feed forward (FF) neural network (NN) with single hidden layer of size 128 as the policy approximator.

The code used for the numerical experiment can be accessed at: https://github.itap.purdue.edu/Clan-labs/NearOptimalLocalPolicy

Before concluding, we would like to point out that both reward and state transition functions considered in our experimental set up are independent of the action distributions. Therefore, the setting described in this section satisfies Assumption 4. Moreover, due to the finiteness of individual state space $\mathcal{Q}$, and action space $\{0,1\}$, the reward function is bounded. Also, it is straightforward to verify the Lipschitz continuity of both reward and transition functions. Hence, Assumptions 1 and 2 are satisfied. Finally, we use neural network (NN) based policies (with bounded weights) in our experiment. Thus, Assumption 3 is also satisfied.

## 9    Conclusions

In this article, we show that, in an $N$-agent system, one can always choose localised policies such that the resulting value function is close to the value function generated by (possibly non-local) optimal policy. We mathematically characterize the approximation error as a function of $N$. Furthermore, we devise an algorithm to explicitly obtain the said local policy. One interesting extension of our problem would be to consider the case where the interactions between the agents are non-uniform. Proving near-optimality of local policies appears to be difficult in such a scenario because, due to the non-uniformity, the notion of mean-field is hard to define. Although our results are pertinent to the standard MARL systems, similar results can also be established for other variants of reinforcement learning problems, e.g., the model described in (Gast and Gaujal, 2011).

**Acknowledgments**

W. U. M. and S. V. U. were partially funded by NSF Grant No. 1638311 CRISP Type 2/Collaborative Research: Critical Transitions in the Resilience and Recovery of Interdependent Social and Physical Networks.

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

# A   Proof of Theorem 1

In order to establish the theorem, the following Lemmas are necessary. The proof of the Lemmas are relegated to Appendix C−I.

## A.1   Lipschitz Continuity Lemmas

In the following three lemmas (Lemma $2-4$), $\pi, \bar{\pi} \in \Pi$ are arbitrary admittable policies, and $\boldsymbol{\mu}, \bar{\boldsymbol{\mu}} \in \mathcal{P}(\mathcal{X})$ are arbitrary state distributions. Moreover, the following definition is frequently used.

$$|\pi(\cdot, \boldsymbol{\mu}) - \bar{\pi}(\cdot, \bar{\boldsymbol{\mu}})|_\infty \triangleq \sup_{x \in \mathcal{X}} |\pi(x, \boldsymbol{\mu}) - \bar{\pi}(x, \bar{\boldsymbol{\mu}})|_1 \tag{28}$$

**Lemma 2.** *If $\nu^{\mathrm{MF}}(\cdot, \cdot)$ is defined by (7), then the following holds.*

$$|\nu^{\mathrm{MF}}(\boldsymbol{\mu}, \pi) - \nu^{\mathrm{MF}}(\bar{\boldsymbol{\mu}}, \bar{\pi})|_1 \leq |\boldsymbol{\mu} - \bar{\boldsymbol{\mu}}|_1 + |\pi(\cdot, \boldsymbol{\mu}) - \bar{\pi}(\cdot, \bar{\boldsymbol{\mu}})|_\infty \tag{29}$$

**Lemma 3.** *If $P^{\mathrm{MF}}(\cdot, \cdot)$ is defined by (8), then the following holds.*

$$|P^{\mathrm{MF}}(\boldsymbol{\mu}, \pi) - P^{\mathrm{MF}}(\bar{\boldsymbol{\mu}}, \bar{\pi})|_1 \leq \tilde{S}_P |\boldsymbol{\mu} - \bar{\boldsymbol{\mu}}|_1 + \bar{S}_P |\pi(\cdot, \boldsymbol{\mu}) - \bar{\pi}(\cdot, \bar{\boldsymbol{\mu}})|_\infty \tag{30}$$

*where $\tilde{S}_P \triangleq 1 + 2L_P$, and $\bar{S}_P \triangleq 1 + L_P$.*

**Lemma 4.** *If $\nu^{\mathrm{MF}}(\cdot, \cdot)$ is defined by (9), then the following holds.*

$$|r^{\mathrm{MF}}(\boldsymbol{\mu}, \pi) - r^{\mathrm{MF}}(\bar{\boldsymbol{\mu}}, \bar{\pi})| \leq \tilde{S}_R |\boldsymbol{\mu} - \bar{\boldsymbol{\mu}}|_1 + \bar{S}_R |\pi(\cdot, \boldsymbol{\mu}) - \bar{\pi}(\cdot, \bar{\boldsymbol{\mu}})|_\infty \tag{31}$$

*where $\tilde{S}_R \triangleq M_R + 2L_R$, and $\bar{S}_R \triangleq M_R + L_R$.*

Lemma $2-4$ essentially dictate that the state and action evolution functions, ($P^{\mathrm{MF}}(\cdot, \cdot)$, and $\nu^{\mathrm{MF}}(\cdot, \cdot)$ respectively), and the average reward function, $r^{\mathrm{MF}}(\cdot, \cdot)$ demonstrate Lipschitz continuity property w. r. t. the state distribution and policy arguments. Lemma 2 is an essential ingredient in the proof of Lemma 3, and 4.

## A.2   Large Population Approximation Lemmas

In the following four lemmas (Lemma $5-8$), $\boldsymbol{\pi} \triangleq \{\pi_t\}_{t \in \{0, 1, \cdots\}} \in \Pi_\infty$ is an arbitrary admissible policy-sequence, and $\{\boldsymbol{\mu}_t^N, \boldsymbol{\nu}_t^N\}_{t \in \{0, 1, \cdots\}}$ are the $N$-agent empirical state, and action distributions induced by it from the initial state distribution, $\boldsymbol{\mu}_0$. Similarly, $\{x_t^i, u_t^i\}_{t \in \{0, 1, \cdots\}}$ are the states, and actions of $i$-th agent evolved from the initial distribution, $\boldsymbol{\mu}_0$ via the policy-sequence, $\boldsymbol{\pi}$. The joint states, and actions at time $t$ are denoted by $\boldsymbol{x}_t^N, \boldsymbol{u}_t^N$ respectively.

**Lemma 5.** *The following inequality holds $\forall t \in \{0, 1, \cdots\}$.*

$$\mathbb{E}\big|\boldsymbol{\nu}_t^N - \nu^{\mathrm{MF}}(\boldsymbol{\mu}_t^N, \pi_t)\big|_1 \leq \frac{1}{\sqrt{N}} \sqrt{|\mathcal{U}|} \tag{32}$$

**Lemma 6.** *The following inequality holds $\forall t \in \{0, 1, \cdots\}$.*

$$\mathbb{E}\big|\boldsymbol{\mu}_{t+1}^N - P^{\mathrm{MF}}(\boldsymbol{\mu}_t^N, \pi_t)\big|_1 \leq \frac{C_P}{\sqrt{N}} \left[\sqrt{|\mathcal{X}|} + \sqrt{|\mathcal{U}|}\right] \tag{33}$$

*where $C_P \triangleq 2 + L_P$.*

**Lemma 7.** *The following inequality holds $\forall t \in \{0, 1, \cdots\}$.*

$$\mathbb{E}\left|\frac{1}{N}\sum_{i=1}^N r(x_t^i, u_t^i, \boldsymbol{\mu}_t^N, \boldsymbol{\nu}_t^N) - r^{\mathrm{MF}}(\boldsymbol{\mu}_t^N, \pi_t)\right| \leq \frac{M_R}{\sqrt{N}} + \frac{L_R}{\sqrt{N}} \sqrt{|\mathcal{U}|}$$

Finally, if $\boldsymbol{\mu}_t^\infty$ indicates the state distribution of an infinite agent system at time $t$ induced by the policy-sequence, $\boldsymbol{\pi}$ from the initial distribution, $\boldsymbol{\mu}_0$ then the following result can be proven invoking Lemma 3, and 6.

**Lemma 8.** *The following inequality holds* $\forall t \in \{0, 1, \cdots\}$.

$$\mathbb{E}|\boldsymbol{\mu}_t^N - \boldsymbol{\mu}_t^\infty| \leq \frac{C_P}{\sqrt{N}} \left[ \sqrt{|\mathcal{X}|} + \sqrt{|\mathcal{U}|} \right] \left( \frac{S_P^t - 1}{S_P - 1} \right)$$

*where* $S_P \triangleq \tilde{S}_P + L_Q \bar{S}_P$. *The terms* $\tilde{S}_P, \bar{S}_P$ *are defined in Lemma 3 while* $C_P$ *is given in Lemma 6.*

### A.3 Proof of the Theorem

Let $\boldsymbol{\pi} \triangleq \{\pi_t\}_{t\in\{0,1,\cdots\}}$, and $\bar{\boldsymbol{\pi}} \triangleq \{\bar{\pi}_t\}_{t\in\{0,1,\cdots\}}$ be two arbitrary policy-sequences in $\Pi_\infty$. Denote by $\{\boldsymbol{\mu}_t^N, \boldsymbol{\nu}_t^N\}, \{\boldsymbol{\mu}_t^\infty, \boldsymbol{\nu}_t^\infty\}$ the state, and action distributions induced by policy-sequence $\boldsymbol{\pi}$ at time $t$ in an $N$-agent system, and infinite agent systems respectively. Also, the state, and action of $i$-th agent at time $t$ corresponding to the same policy sequence are indicated as $x_t^i$, and $u_t^i$ respectively. The same quantities corresponding to the policy-sequence $\bar{\boldsymbol{\pi}}$ are denoted as $\{\bar{\boldsymbol{\mu}}_t^N, \bar{\boldsymbol{\nu}}_t^N, \bar{\boldsymbol{\mu}}_t^\infty, \bar{\boldsymbol{\nu}}_t^\infty, \bar{x}_t^i, \bar{u}_t^i\}$. Consider the following difference,

$$|v_{\text{MARL}}(\boldsymbol{x}_0^N, \bar{\boldsymbol{\pi}}) - v_{\text{MF}}(\boldsymbol{\mu}_0, \boldsymbol{\pi})|$$

$$\overset{(a)}{\leq} \sum_{t=0}^\infty \gamma^t \left| \frac{1}{N} \sum_{i=1}^N \mathbb{E}\left[ r(\bar{x}_t^i, \bar{u}_t^i, \bar{\boldsymbol{\mu}}_t^N, \bar{\boldsymbol{\nu}}_t^N) \right] - r^{\text{MF}}(\boldsymbol{\mu}_t^\infty, \pi_t) \right|$$

$$\leq \underbrace{\sum_{t=0}^\infty \gamma^t \mathbb{E} \left| \frac{1}{N} r(\bar{x}_t^i, \bar{u}_t^i, \bar{\boldsymbol{\mu}}_t^N, \bar{\boldsymbol{\nu}}_t^N) - r^{\text{MF}}(\bar{\boldsymbol{\mu}}_t^N, \bar{\pi}_t) \right|}_{\triangleq J_1} + \underbrace{\sum_{t=0}^\infty \gamma^t \mathbb{E} \left| r^{\text{MF}}(\bar{\boldsymbol{\mu}}_t^N, \bar{\pi}_t) - r^{\text{MF}}(\bar{\boldsymbol{\mu}}_t^\infty, \bar{\pi}_t) \right|}_{\triangleq J_2}$$

$$+ \underbrace{\sum_{t=0}^\infty \gamma^t \left| r^{\text{MF}}(\bar{\boldsymbol{\mu}}_t^\infty, \bar{\pi}_t) - r^{\text{MF}}(\boldsymbol{\mu}_t^\infty, \pi_t) \right|}_{\triangleq J_3}$$

Inequality (a) follows from the definition of the value functions $v_{\text{MARL}}(\cdot, \cdot)$, $v_{\text{MF}}(\cdot, \cdot)$ given in (6), and (10) respectively. The first term, $J_1$ can be bounded using Lemma 7 as follows.

$$J_1 \leq \left( \frac{1}{1-\gamma} \right) \left[ \frac{M_R}{\sqrt{N}} + \frac{L_R}{\sqrt{N}} \sqrt{|\mathcal{U}|} \right]$$

The second term, $J_2$, can be bounded as follows.

$$J_2 \triangleq \sum_{t=0}^\infty \gamma^t \mathbb{E} |r^{\text{MF}}(\bar{\boldsymbol{\mu}}_t^N, \bar{\pi}_t) - r^{\text{MF}}(\bar{\boldsymbol{\mu}}_t^\infty, \bar{\pi}_t)|$$

$$\overset{(a)}{\leq} \sum_{t=0}^\infty \gamma^t \mathbb{E} \left\{ \tilde{S}_R |\bar{\boldsymbol{\mu}}_t^N - \bar{\boldsymbol{\mu}}_t^\infty|_1 + \bar{S}_R \left| \bar{\pi}_t(\cdot, \bar{\boldsymbol{\mu}}_t^N) - \bar{\pi}_t(\cdot, \bar{\boldsymbol{\mu}}_t^\infty) \right|_\infty \right\}$$

$$\overset{(b)}{\leq} S_R \sum_{t=0}^\infty \gamma^t \mathbb{E} |\bar{\boldsymbol{\mu}}_t^N - \bar{\boldsymbol{\mu}}_t^\infty|$$

$$\overset{(c)}{\leq} \frac{1}{\sqrt{N}} \left[ \sqrt{|\mathcal{X}|} + \sqrt{|\mathcal{U}|} \right] \left( \frac{S_R C_P}{S_P - 1} \right) \left[ \frac{1}{1 - \gamma S_P} - \frac{1}{1 - \gamma} \right]$$

where $S_R \triangleq \tilde{S}_R + L_Q \bar{S}_R$. Inequality (a) follows from Lemma 4, whereas (b) is a consequence of Assumption 3. Finally, (c) follows from Lemma 8. It remains to bound $J_3$. Note that, if $\bar{\boldsymbol{\pi}} = \boldsymbol{\pi}$, then $J_3 = 0$. Hence,

$$|v_{\text{MARL}}(\boldsymbol{x}_0^N, \boldsymbol{\pi}_{\text{MARL}}^*) - v_{\text{MF}}(\boldsymbol{\mu}_0, \boldsymbol{\pi}_{\text{MARL}}^*)| \leq J_0, \tag{34}$$

$$|v_{\text{MARL}}(\boldsymbol{x}_0^N, \boldsymbol{\pi}_{\text{MF}}^*) - v_{\text{MF}}(\boldsymbol{\mu}_0, \boldsymbol{\pi}_{\text{MF}}^*)| \leq J_0 \tag{35}$$

where $J_0$ is given as follows,

$$J_0 \triangleq \left(\frac{1}{1-\gamma}\right)\left[\frac{M_R}{\sqrt{N}} + \frac{L_R}{\sqrt{N}}\sqrt{|\mathcal{U}|}\right] + \frac{1}{\sqrt{N}}\left[\sqrt{|\mathcal{X}|} + \sqrt{|\mathcal{U}|}\right]\left(\frac{S_R C_P}{S_P - 1}\right)\left[\frac{1}{1-\gamma S_P} - \frac{1}{1-\gamma}\right]$$

Moreover, if $\boldsymbol{\pi} = \boldsymbol{\pi}_{\mathrm{MF}}^*$, and $\bar{\boldsymbol{\pi}} = \tilde{\boldsymbol{\pi}}_{\mathrm{MF}}^*$ (or vice versa), then $J_3 = 0$ as well. This is precisely because the trajectory of state, and action distributions generated by the policy-sequences $\boldsymbol{\pi}_{\mathrm{MF}}^*, \tilde{\boldsymbol{\pi}}_{\mathrm{MF}}^*$ are identical in an infinite agent system. Hence, we have,

$$|v_{\mathrm{MARL}}(\boldsymbol{x}_0^N, \tilde{\boldsymbol{\pi}}_{\mathrm{MF}}^*) - v_{\mathrm{MF}}(\boldsymbol{\mu}_0, \boldsymbol{\pi}_{\mathrm{MF}}^*)| \leq J_0 \tag{36}$$

Consider the following inequalities,

$$\begin{aligned}
&v_{\mathrm{MARL}}(\boldsymbol{x}_0^N, \boldsymbol{\pi}_{\mathrm{MARL}}^*) - v_{\mathrm{MARL}}(\boldsymbol{x}_0^N, \tilde{\boldsymbol{\pi}}_{\mathrm{MF}}^*) \\
&= v_{\mathrm{MARL}}(\boldsymbol{x}_0^N, \boldsymbol{\pi}_{\mathrm{MARL}}^*) - v_{\mathrm{MF}}(\boldsymbol{\mu}_0, \boldsymbol{\pi}_{\mathrm{MF}}^*) + v_{\mathrm{MF}}(\boldsymbol{\mu}_0, \boldsymbol{\pi}_{\mathrm{MF}}^*) - v_{\mathrm{MARL}}(\boldsymbol{x}_0^N, \tilde{\boldsymbol{\pi}}_{\mathrm{MF}}^*) \\
&\stackrel{(a)}{\leq} v_{\mathrm{MARL}}(\boldsymbol{x}_0^N, \boldsymbol{\pi}_{\mathrm{MARL}}^*) - v_{\mathrm{MF}}(\boldsymbol{\mu}_0, \boldsymbol{\pi}_{\mathrm{MARL}}^*) + J_0 \stackrel{(b)}{\leq} 2J_0
\end{aligned} \tag{37}$$

Inequality (a) follows from (36), and the fact that $\boldsymbol{\pi}_{\mathrm{MF}}^*$ maximizes $v_{\mathrm{MF}}(\boldsymbol{\mu}_0, \cdot)$. Inequality (b) follows from (34). Moreover,

$$\begin{aligned}
&v_{\mathrm{MARL}}(\boldsymbol{x}_0^N, \tilde{\boldsymbol{\pi}}_{\mathrm{MF}}^*) - v_{\mathrm{MARL}}(\boldsymbol{x}_0^N, \boldsymbol{\pi}_{\mathrm{MARL}}^*) \\
&= v_{\mathrm{MARL}}(\boldsymbol{x}_0^N, \tilde{\boldsymbol{\pi}}_{\mathrm{MF}}^*) - v_{\mathrm{MF}}(\boldsymbol{\mu}_0, \boldsymbol{\pi}_{\mathrm{MF}}^*) + v_{\mathrm{MF}}(\boldsymbol{\mu}_0, \boldsymbol{\pi}_{\mathrm{MF}}^*) - v_{\mathrm{MARL}}(\boldsymbol{x}_0^N, \boldsymbol{\pi}_{\mathrm{MARL}}^*) \\
&\stackrel{(a)}{\leq} J_0 + v_{\mathrm{MF}}(\boldsymbol{\mu}_0, \boldsymbol{\pi}_{\mathrm{MF}}^*) - v_{\mathrm{MARL}}(\boldsymbol{x}_0^N, \boldsymbol{\pi}_{\mathrm{MF}}^*) \stackrel{(b)}{\leq} 2J_0
\end{aligned} \tag{38}$$

Inequality (a) follows from (36), and the fact that $\boldsymbol{\pi}_{\mathrm{MARL}}^*$ maximizes $v_{\mathrm{MARL}}(\boldsymbol{x}_0, \cdot)$. Inequality (b) follows from (35). Combining (37), and (38), we conclude that,

$$|v_{\mathrm{MARL}}(\boldsymbol{x}_0^N, \boldsymbol{\pi}_{\mathrm{MARL}}^*) - v_{\mathrm{MARL}}(\boldsymbol{x}_0^N, \tilde{\boldsymbol{\pi}}_{\mathrm{MF}}^*)| \leq 2J_0$$

## B  Proof of Theorem 2

The proof of Theorem 2 is similar to that of Theorem 1, however, with subtle differences. The following Lemmas are needed to establish the theorem.

### B.1  Auxiliary Lemmas

In the following (Lemma $9 - 11$), $\boldsymbol{\pi} \triangleq \{\pi_t\}_{t \in \{0,1,\cdots\}} \in \Pi_\infty$ denotes an arbitrary admissible policy-sequence. The terms $\{\boldsymbol{\mu}_t^N, \boldsymbol{\nu}_t^N\}_{t \in \{0,1,\cdots\}}$ denote the $N$-agent empirical state, and action distributions induced by $\boldsymbol{\pi}$ from the initial state distribution, $\boldsymbol{\mu}_0$ whereas $\{\boldsymbol{\mu}_t^\infty, \boldsymbol{\nu}_t^\infty\}_{t \in \{0,1,\cdots\}}$ indicate the state, and action distributions induced by the same policy-sequence in an infinite agent system. Similarly, $\{x_t^i, u_t^i\}_{t \in \{0,1,\cdots\}}$ denote the states, and actions of $i$-th agent evolved from the initial distribution, $\boldsymbol{\mu}_0$ via the policy-sequence, $\boldsymbol{\pi}$. The joint states, and actions at time $t$ are denoted by $\boldsymbol{x}_t^N, \boldsymbol{u}_t^N$ respectively.

**Lemma 9.** *The following inequality holds* $\forall t \in \{0, 1, \cdots\}$.

$$\mathbb{E}\left|\boldsymbol{\mu}_{t+1}^N - P^{\mathrm{MF}}(\boldsymbol{\mu}_t^N, \pi_t)\right|_1 \leq \frac{2}{\sqrt{N}}\sqrt{|\mathcal{X}|} \tag{39}$$

**Lemma 10.** *The following inequality holds* $\forall t \in \{0, 1, \cdots\}$.

$$\mathbb{E}\left|\frac{1}{N}\sum_{i=1}^N r(x_t^i, u_t^i, \boldsymbol{\mu}_t^N) - r^{\mathrm{MF}}(\boldsymbol{\mu}_t^N, \pi_t)\right| \leq \frac{M_R}{\sqrt{N}}$$

**Lemma 11.** *The following inequality holds* $\forall t \in \{0, 1, \cdots\}$.

$$\mathbb{E}|\boldsymbol{\mu}_t^N - \boldsymbol{\mu}_t^\infty| \le \frac{2}{\sqrt{N}}\sqrt{|\mathcal{X}|}\left(\frac{S_P^t - 1}{S_P - 1}\right)$$

*where* $S_P \triangleq \tilde{S}_P + L_Q \bar{S}_P$. *The terms* $\tilde{S}_P, \bar{S}_P$ *are defined in Lemma 3.*

The proofs of the above lemmas are relegated to Appendix $J - L$.

## B.2 Proof of the Theorem

We use the same notations as in Appendix A.3. Consider the following difference,

$$
\begin{aligned}
&|v_{\text{MARL}}(\boldsymbol{x}_0^N, \bar{\boldsymbol{\pi}}) - v_{\text{MF}}(\boldsymbol{\mu}_0, \boldsymbol{\pi})| \\
&\overset{(a)}{\le} \sum_{t=0}^\infty \gamma^t \left| \frac{1}{N}\sum_{i=1}^N r(\bar{x}_t^i, \bar{u}_t^i, \bar{\boldsymbol{\mu}}_t^N) \right] - \mathbb{E}\left[r^{\text{MF}}(\boldsymbol{\mu}_t^\infty, \pi_t)\right| \\
&\le \underbrace{\sum_{t=0}^\infty \gamma^t \mathbb{E}\left|\frac{1}{N}r(\bar{x}_t^i, \bar{u}_t^i, \bar{\boldsymbol{\mu}}_t^N) - r^{\text{MF}}(\bar{\boldsymbol{\mu}}_t^N, \bar{\pi}_t)\right|}_{\triangleq J_1} + \underbrace{\sum_{t=0}^\infty \gamma^t \mathbb{E}\left|r^{\text{MF}}(\bar{\boldsymbol{\mu}}_t^N, \bar{\pi}_t) - r^{\text{MF}}(\bar{\boldsymbol{\mu}}_t^\infty, \bar{\pi}_t)\right|}_{\triangleq J_2} \\
&+ \underbrace{\sum_{t=0}^\infty \gamma^t \left|r^{\text{MF}}(\bar{\boldsymbol{\mu}}_t^\infty, \bar{\pi}_t) - r^{\text{MF}}(\boldsymbol{\mu}_t^\infty, \pi_t)\right|}_{\triangleq J_3}
\end{aligned}
$$

Inequality (a) follows from the definition of the value functions $v_{\text{MARL}}(\cdot, \cdot)$, $v_{\text{MF}}(\cdot, \cdot)$ given in (6), and (10) respectively. The first term, $J_1$ can be bounded using Lemma 7 as follows.

$$J_1 \le \left(\frac{1}{1 - \gamma}\right)\left[\frac{M_R}{\sqrt{N}}\right]$$

The second term, $J_2$, can be bounded as follows.

$$
\begin{aligned}
J_2 &\triangleq \sum_{t=0}^\infty \gamma^t \mathbb{E}|r^{\text{MF}}(\bar{\boldsymbol{\mu}}_t^N, \bar{\pi}_t) - r^{\text{MF}}(\bar{\boldsymbol{\mu}}_t^\infty, \bar{\pi}_t)| \\
&\overset{(a)}{\le} \sum_{t=0}^\infty \gamma^t \mathbb{E}\left\{\tilde{S}_R|\bar{\boldsymbol{\mu}}_t^N - \bar{\boldsymbol{\mu}}_t^\infty|_1 + \bar{S}_R\left|\bar{\pi}_t(\cdot, \bar{\boldsymbol{\mu}}_t^N) - \bar{\pi}_t(\cdot, \bar{\boldsymbol{\mu}}_t^\infty)\right|_\infty\right\} \\
&\overset{(b)}{\le} S_R \sum_{t=0}^\infty \gamma^t \mathbb{E}|\bar{\boldsymbol{\mu}}_t^N - \bar{\boldsymbol{\mu}}_t^\infty| \\
&\overset{(c)}{\le} \frac{1}{\sqrt{N}}\sqrt{|\mathcal{X}|}\left(\frac{2S_R}{S_P - 1}\right)\left[\frac{1}{1 - \gamma S_P} - \frac{1}{1 - \gamma}\right]
\end{aligned}
$$

where $S_R \triangleq \tilde{S}_R + L_Q \bar{S}_R$. Inequality (a) follows from Lemma 4, whereas (b) is a consequence of Assumption 3. Finally, (c) follows from Lemma 8. It remains to bound $J_3$. Note that, if $\bar{\boldsymbol{\pi}} = \boldsymbol{\pi}$, then $J_3 = 0$. Hence,

$$|v_{\text{MARL}}(\boldsymbol{x}_0^N, \boldsymbol{\pi}_{\text{MARL}}^*) - v_{\text{MF}}(\boldsymbol{\mu}_0, \boldsymbol{\pi}_{\text{MARL}}^*)| \le J_0, \tag{40}$$

$$|v_{\text{MARL}}(\boldsymbol{x}_0^N, \boldsymbol{\pi}_{\text{MF}}^*) - v_{\text{MF}}(\boldsymbol{\mu}_0, \boldsymbol{\pi}_{\text{MF}}^*)| \le J_0 \tag{41}$$

where $J_0$ is given as follows,

$$J_0 \triangleq \left(\frac{1}{1 - \gamma}\right)\left[\frac{M_R}{\sqrt{N}}\right] + \frac{1}{\sqrt{N}}\left[\sqrt{|\mathcal{X}|} + \sqrt{|\mathcal{U}|}\right]\left(\frac{2S_R}{S_P - 1}\right)\left[\frac{1}{1 - \gamma S_P} - \frac{1}{1 - \gamma}\right]$$

Moreover, if $\boldsymbol{\pi} = \boldsymbol{\pi}^*_{\mathrm{MF}}$, and $\bar{\boldsymbol{\pi}} = \tilde{\boldsymbol{\pi}}^*_{\mathrm{MF}}$ (or vice versa), then $J_3 = 0$ as well. This is precisely because the trajectory of state, and action distributions generated by the policy-sequences $\boldsymbol{\pi}^*_{\mathrm{MF}}, \tilde{\boldsymbol{\pi}}^*_{\mathrm{MF}}$ are identical in an infinite agent system. Hence, we have,

$$|v_{\mathrm{MARL}}(\boldsymbol{x}_0^N, \tilde{\boldsymbol{\pi}}^*_{\mathrm{MF}}) - v_{\mathrm{MF}}(\boldsymbol{\mu}_0, \boldsymbol{\pi}^*_{\mathrm{MF}})| \le J_0 \tag{42}$$

Following the same set of arguments as is used in (37), and (38), we conclude that,

$$|v_{\mathrm{MARL}}(\boldsymbol{x}_0^N, \boldsymbol{\pi}^*_{\mathrm{MARL}}) - v_{\mathrm{MARL}}(\boldsymbol{\mu}_0, \tilde{\boldsymbol{\pi}}^*_{\mathrm{MF}})| \le 2J_0$$

## C  Proof of Lemma 2

Note the chain of inequalities stated below.

$$
\begin{aligned}
&|\nu^{\mathrm{MF}}(\boldsymbol{\mu}, \pi) - \nu^{\mathrm{MF}}(\bar{\boldsymbol{\mu}}, \bar{\pi})|_1 \\
&\stackrel{(a)}{=} \left| \sum_{x \in \mathcal{X}} \pi(x, \boldsymbol{\mu})\boldsymbol{\mu}(x) - \sum_{x \in \mathcal{X}} \bar{\pi}(x, \bar{\boldsymbol{\mu}})\bar{\boldsymbol{\mu}}(x) \right|_1 \\
&= \sum_{u \in \mathcal{U}} \left| \sum_{x \in \mathcal{X}} \pi(x, \boldsymbol{\mu})(u)\boldsymbol{\mu}(x) - \sum_{x \in \mathcal{X}} \bar{\pi}(x, \bar{\boldsymbol{\mu}})(u)\bar{\boldsymbol{\mu}}(x) \right| \\
&\le \sum_{x \in \mathcal{X}} \sum_{u \in \mathcal{U}} |\pi(x, \boldsymbol{\mu})(u)\boldsymbol{\mu}(x) - \bar{\pi}(x, \bar{\boldsymbol{\mu}})(u)\bar{\boldsymbol{\mu}}(x)| \\
&\le \sum_{x \in \mathcal{X}} |\boldsymbol{\mu}(x) - \bar{\boldsymbol{\mu}}(x)| \underbrace{\sum_{u \in \mathcal{U}} \pi(x, \boldsymbol{\mu})(u)}_{=1} + \sum_{x \in \mathcal{X}} \bar{\boldsymbol{\mu}}(x) \sum_{u \in \mathcal{U}} |\pi(x, \boldsymbol{\mu})(u) - \bar{\pi}(x, \bar{\boldsymbol{\mu}})(u)| \\
&\le |\boldsymbol{\mu} - \bar{\boldsymbol{\mu}}|_1 + \underbrace{\sum_{x \in \mathcal{X}} \bar{\boldsymbol{\mu}}(x)}_{=1} \left[ \sup_{x \in \mathcal{X}} |\pi(x, \boldsymbol{\mu}) - \bar{\pi}(x, \bar{\boldsymbol{\mu}})|_1 \right] \\
&\stackrel{(b)}{=} |\boldsymbol{\mu} - \bar{\boldsymbol{\mu}}|_1 + |\pi(\cdot, \boldsymbol{\mu}) - \bar{\pi}(\cdot, \bar{\boldsymbol{\mu}})|_\infty
\end{aligned}
$$

Inequality (a) follows from the definition of $\nu^{\mathrm{MF}}(\cdot, \cdot)$ as given in (7). On the other hand, equation (b) is a consequence of the definition of $|\cdot|_\infty$ over the space of all admissible policies, $\Pi$. This concludes the result.

## D  Proof of Lemma 3

Observe that,

$$
\begin{aligned}
&|P^{\mathrm{MF}}(\boldsymbol{\mu}, \pi) - P^{\mathrm{MF}}(\bar{\boldsymbol{\mu}}, \bar{\pi})|_1 \\
&\stackrel{(a)}{=} \left| \sum_{x \in \mathcal{X}} \sum_{u \in \mathcal{U}} P(x, u, \boldsymbol{\mu}, \nu^{\mathrm{MF}}(\boldsymbol{\mu}, \pi))\pi(x, \boldsymbol{\mu})(u)\boldsymbol{\mu}(x) - P(x, u, \bar{\boldsymbol{\mu}}, \nu^{\mathrm{MF}}(\bar{\boldsymbol{\mu}}, \bar{\pi}))\bar{\pi}(x, \bar{\boldsymbol{\mu}})(u)\bar{\boldsymbol{\mu}}(x) \right|_1 \\
&\le J_1 + J_2
\end{aligned}
$$

Equality (a) follows from the definition of $P^{\mathrm{MF}}(\cdot, \cdot)$ as depicted in (8). The term $J_1$ satisfies the following bound.

$$
\begin{aligned}
J_1 &\triangleq \sum_{x \in \mathcal{X}} \sum_{u \in \mathcal{U}} \left| P(x, u, \boldsymbol{\mu}, \nu^{\mathrm{MF}}(\boldsymbol{\mu}, \pi)) - P(x, u, \bar{\boldsymbol{\mu}}, \nu^{\mathrm{MF}}(\bar{\boldsymbol{\mu}}, \bar{\pi})) \right|_1 \times \pi(x, \boldsymbol{\mu})(u) \boldsymbol{\mu}(x) \\
&\stackrel{(a)}{\leq} L_P \left[ |\boldsymbol{\mu} - \bar{\boldsymbol{\mu}}|_1 + |\nu^{\mathrm{MF}}(\boldsymbol{\mu}, \pi) - \nu^{\mathrm{MF}}(\bar{\boldsymbol{\mu}}, \bar{\pi})|_1 \right] \times \underbrace{\sum_{x \in \mathcal{X}} \boldsymbol{\mu}(x) \sum_{u \in \mathcal{U}} \pi(x, \boldsymbol{\mu})(u)}_{=1} \\
&\stackrel{(b)}{\leq} 2 L_P |\boldsymbol{\mu} - \bar{\boldsymbol{\mu}}|_1 + L_P |\pi(\cdot, \boldsymbol{\mu}) - \bar{\pi}(\cdot, \bar{\boldsymbol{\mu}})|_\infty
\end{aligned}
$$

Inequality $(a)$ is a consequence of Assumption 2 whereas $(b)$ follows from Lemma 2, and the fact that $\pi(x, \boldsymbol{\mu})$, $\boldsymbol{\mu}$ are probability distributions. The second term, $J_2$ obeys the following bound.

$$
\begin{aligned}
J_2 &\triangleq \sum_{x \in \mathcal{X}} \sum_{u \in \mathcal{U}} \underbrace{|P(x, u, \bar{\boldsymbol{\mu}}, \nu^{\mathrm{MF}}(\bar{\boldsymbol{\mu}}, \bar{\pi}))|_1}_{=1} \times |\pi(x, \boldsymbol{\mu})(u) \boldsymbol{\mu}(x) - \bar{\pi}(x, \bar{\boldsymbol{\mu}})(u) \bar{\boldsymbol{\mu}}(x)| \\
&\leq \sum_{x \in \mathcal{X}} |\boldsymbol{\mu}(x) - \bar{\boldsymbol{\mu}}(x)| \underbrace{\sum_{u \in \mathcal{U}} \pi(x, \boldsymbol{\mu})(u)}_{=1} + \sum_{x \in \mathcal{X}} \bar{\boldsymbol{\mu}}(x) \sum_{u \in \mathcal{U}} |\pi(x, \boldsymbol{\mu})(u) - \bar{\pi}(x, \bar{\boldsymbol{\mu}})(u)| \\
&\stackrel{(a)}{\leq} |\boldsymbol{\mu} - \bar{\boldsymbol{\mu}}|_1 + \underbrace{\sum_{x \in \mathcal{X}} \bar{\boldsymbol{\mu}}(x)}_{=1} \left[ \sup_{x \in \mathcal{X}} |\pi(x, \boldsymbol{\mu}) - \bar{\pi}(x, \bar{\boldsymbol{\mu}})|_1 \right] \\
&= |\boldsymbol{\mu} - \bar{\boldsymbol{\mu}}|_1 + |\pi(\cdot, \boldsymbol{\mu}) - \bar{\pi}(\cdot, \bar{\boldsymbol{\mu}})|_\infty
\end{aligned}
$$

Inequality $(a)$ results from the fact that $\pi(x, \boldsymbol{\mu})$ is a probability distribution while $(b)$ utilizes the definition of $|\cdot|_\infty$. This concludes the result.

## E   Proof of Lemma 4

Observe that,

$$
\begin{aligned}
&|r^{\mathrm{MF}}(\boldsymbol{\mu}, \pi) - r^{\mathrm{MF}}(\bar{\boldsymbol{\mu}}, \bar{\pi})| \\
&\stackrel{(a)}{=} \left| \sum_{x \in \mathcal{X}} \sum_{u \in \mathcal{U}} r(x, u, \boldsymbol{\mu}, \nu^{\mathrm{MF}}(\boldsymbol{\mu}, \pi)) \pi(x, \boldsymbol{\mu})(u) \boldsymbol{\mu}(x) - r(x, u, \bar{\boldsymbol{\mu}}, \nu^{\mathrm{MF}}(\bar{\boldsymbol{\mu}}, \bar{\pi})) \bar{\pi}(x, \bar{\boldsymbol{\mu}})(u) \bar{\boldsymbol{\mu}}(x) \right| \\
&\leq J_1 + J_2
\end{aligned}
$$

Equality (a) follows from the definition of $r^{\mathrm{MF}}(\cdot, \cdot)$ as given in (9). The first term obeys the following bound.

$$
\begin{aligned}
J_1 &\triangleq \sum_{x \in \mathcal{X}} \sum_{u \in \mathcal{U}} \left| r(x, u, \boldsymbol{\mu}, \nu^{\mathrm{MF}}(\boldsymbol{\mu}, \pi)) - r(x, u, \bar{\boldsymbol{\mu}}, \nu^{\mathrm{MF}}(\bar{\boldsymbol{\mu}}, \bar{\pi})) \right| \times \pi(x, \boldsymbol{\mu})(u) \boldsymbol{\mu}(x) \\
&\stackrel{(a)}{\leq} L_R \left[ |\boldsymbol{\mu} - \bar{\boldsymbol{\mu}}|_1 + |\nu^{\mathrm{MF}}(\boldsymbol{\mu}, \pi) - \nu^{\mathrm{MF}}(\bar{\boldsymbol{\mu}}, \bar{\pi})|_1 \right] \times \underbrace{\sum_{x \in \mathcal{X}} \boldsymbol{\mu}(x) \sum_{u \in \mathcal{U}} \pi(x, \boldsymbol{\mu})(u)}_{=1} \\
&\stackrel{(b)}{\leq} 2 L_R |\boldsymbol{\mu} - \bar{\boldsymbol{\mu}}|_1 + L_R |\pi(\cdot, \boldsymbol{\mu}) - \bar{\pi}(\cdot, \bar{\boldsymbol{\mu}})|_\infty
\end{aligned}
$$

Inequality ($a$) is a consequence of Assumption 1(b) whereas inequality ($b$) follows from Lemma 2, and the fact that $\pi(x, \boldsymbol{\mu})$, $\boldsymbol{\mu}$ are probability distributions. The second term, $J_2$ satisfies the following.

$$
\begin{aligned}
J_2 &\triangleq \sum_{x \in \mathcal{X}} \sum_{u \in \mathcal{U}} |r(x, u, \bar{\boldsymbol{\mu}}, \nu^{\mathrm{MF}}(\bar{\boldsymbol{\mu}}, \bar{\pi}))| \times |\pi(x, \boldsymbol{\mu})(u)\boldsymbol{\mu}(x) - \bar{\pi}(x, \bar{\boldsymbol{\mu}})(u)\bar{\boldsymbol{\mu}}(x)| \\
&\overset{(a)}{\leq} M_R \sum_{x \in \mathcal{X}} \sum_{u \in \mathcal{U}} |\pi(x, \boldsymbol{\mu})(u)\boldsymbol{\mu}(x) - \bar{\pi}(x, \bar{\boldsymbol{\mu}})(u)\bar{\boldsymbol{\mu}}(x)| \\
&\leq M_R \sum_{x \in \mathcal{X}} |\boldsymbol{\mu}(x) - \bar{\boldsymbol{\mu}}(x)| \underbrace{\sum_{u \in \mathcal{U}} \pi(x, \boldsymbol{\mu})(u)}_{=1} + M_R \sum_{x \in \mathcal{X}} \bar{\boldsymbol{\mu}}(x) \sum_{u \in \mathcal{U}} |\pi(x, \boldsymbol{\mu})(u) - \bar{\pi}(x, \bar{\boldsymbol{\mu}})(u)| \\
&\leq M_R |\boldsymbol{\mu} - \bar{\boldsymbol{\mu}}|_1 + M_R \underbrace{\sum_{x \in \mathcal{X}} \bar{\boldsymbol{\mu}}(x)}_{=1} \left[ \sup_{x \in \mathcal{X}} |\pi(x, \boldsymbol{\mu}) - \bar{\pi}(x, \bar{\boldsymbol{\mu}})|_1 \right] \\
&\overset{(b)}{=} M_R |\boldsymbol{\mu} - \bar{\boldsymbol{\mu}}|_1 + M_R |\pi(\cdot, \boldsymbol{\mu}) - \bar{\pi}(\cdot, \bar{\boldsymbol{\mu}})|_\infty
\end{aligned}
$$

Inequality ($a$) results from Assumption 1(a). On the other hand, equality ($b$) is a consequence of the definition of $|\cdot|_\infty$, and the fact that $\bar{\boldsymbol{\mu}}$ is a probability distribution. This concludes the result.

## F  Proof of Lemma 5

The following Lemma is required to prove the result.

**Lemma 12.** *If $\forall m \in \{1, \cdots, M\}$, $\{X_{mn}\}_{n \in \{1, \cdots, N\}}$ are independent random variables that lie in $[0, 1]$, and satisfy $\sum_{m \in \{1, \cdots, M\}} \mathbb{E}[X_{mn}] \leq 1$, $\forall n \in \{1, \cdots, N\}$, then the following holds,*

$$
\sum_{m=1}^{M} \mathbb{E} \left| \sum_{n=1}^{N} (X_{mn} - \mathbb{E}[X_{mn}]) \right| \leq \sqrt{MN} \tag{43}
$$

Lemma 12 is adapted from Lemma 13 of (Mondal et al., 2022a).

Notice the following relations.

$$
\begin{aligned}
&\mathbb{E} \left| \boldsymbol{\nu}_t^N - \nu^{\mathrm{MF}}(\boldsymbol{\mu}_t^N, \pi_t) \right|_1 \\
&= \mathbb{E} \left[ \mathbb{E} \left[ \left| \boldsymbol{\nu}_t^N - \nu^{\mathrm{MF}}(\boldsymbol{\mu}_t^N, \pi_t) \right|_1 \Big| \boldsymbol{x}_t^N \right] \right] \\
&\overset{(a)}{=} \mathbb{E} \left[ \mathbb{E} \left[ \left| \boldsymbol{\nu}_t^N - \sum_{x \in \mathcal{X}} \pi_t(x, \boldsymbol{\mu}_t^N)\boldsymbol{\mu}_t^N(x) \right|_1 \Big| \boldsymbol{x}_t^N \right] \right] \\
&= \mathbb{E} \left[ \mathbb{E} \left[ \sum_{u \in \mathcal{U}} \left| \boldsymbol{\nu}_t^N(u) - \sum_{x \in \mathcal{X}} \pi_t(x, \boldsymbol{\mu}_t^N)(u)\boldsymbol{\mu}_t^N(x) \right| \Big| \boldsymbol{x}_t^N \right] \right] \\
&\overset{(b)}{=} \mathbb{E} \left[ \sum_{u \in \mathcal{U}} \mathbb{E} \left[ \frac{1}{N} \left| \sum_{i=1}^{N} \delta(u_t^i = u) - \frac{1}{N} \sum_{x \in \mathcal{X}} \pi_t(x, \boldsymbol{\mu}_t^N)(u) \sum_{i=1}^{N} \delta(x_t^i = x) \right| \Big| \boldsymbol{x}_t^N \right] \right] \\
&= \mathbb{E} \left[ \sum_{u \in \mathcal{U}} \mathbb{E} \left[ \left| \frac{1}{N} \sum_{i=1}^{N} \delta(u_t^i = u) - \frac{1}{N} \sum_{i=1}^{N} \pi_t(x_t^i, \boldsymbol{\mu}_t^N)(u) \right| \Big| \boldsymbol{x}_t^N \right] \right] \\
&\overset{(c)}{\leq} \frac{1}{\sqrt{N}} \sqrt{|\mathcal{U}|}
\end{aligned}
$$

Equality (a) can be established using the definition of $\nu^{\mathrm{MF}}(\cdot, \cdot)$ as depicted in (7). Similarly, relation (b) is a consequence of the definitions of $\boldsymbol{\mu}_t^N, \boldsymbol{\nu}_t^N$. Finally, (c) uses Lemma 12. Specifically, it utilises the facts that,

$\{u_t^i\}_{i \in \{1,\cdots,N\}}$ are conditionally independent given $\boldsymbol{x}_t^N$, and the followings hold

$$\mathbb{E}\left[\delta(u_t^i = u)\Big|\boldsymbol{x}_t^N\right] = \pi_t(x_t^i, \boldsymbol{\mu}_t^N)(u),$$

$$\sum_{u \in \mathcal{U}} \mathbb{E}\left[\delta(u_t^i = u)\Big|\boldsymbol{x}_t^N\right] = 1$$

$\forall i \in \{1,\cdots,N\}, \forall u \in \mathcal{U}$. This concludes the lemma.

## G  Proof of Lemma 6

Notice the following decomposition.

$$\mathbb{E}\left|\boldsymbol{\mu}_{t+1}^N - P^{\mathrm{MF}}(\boldsymbol{\mu}_t^N, \pi_t)\right|_1$$

$$\overset{(a)}{=} \sum_{x \in \mathcal{X}} \mathbb{E}\left|\frac{1}{N}\sum_{i=1}^N \delta(x_{t+1}^i = x) - \sum_{x' \in \mathcal{X}}\sum_{u \in \mathcal{U}} P(x', u, \boldsymbol{\mu}_t^N, \nu^{\mathrm{MF}}(\boldsymbol{\mu}_t^N, \pi_t))(x)\pi_t(x', \boldsymbol{\mu}_t^N)(u)\frac{1}{N}\sum_{i=1}^N \delta(x_t^i = x')\right|$$

$$= \sum_{x \in \mathcal{X}} \mathbb{E}\left|\frac{1}{N}\sum_{i=1}^N \delta(x_{t+1}^i = x) - \frac{1}{N}\sum_{i=1}^N\sum_{u \in \mathcal{U}} P(x_t^i, u, \boldsymbol{\mu}_t^N, \nu^{\mathrm{MF}}(\boldsymbol{\mu}_t^N, \pi_t))(x)\pi_t(x_t^i, \boldsymbol{\mu}_t^N)(u)\right|$$

$$\leq J_1 + J_2 + J_3$$

Equality (a) uses the definition of $P^{\mathrm{MF}}(\cdot, \cdot)$ as shown in (8). The term, $J_1$ obeys the following bound.

$$J_1 \triangleq \frac{1}{N}\sum_{x \in \mathcal{X}} \mathbb{E}\left|\sum_{i=1}^N \delta(x_{t+1}^i = x) - \sum_{i=1}^N P(x_t^i, u_t^i, \boldsymbol{\mu}_t^N, \boldsymbol{\nu}_t^N)(x)\right|$$

$$= \frac{1}{N}\sum_{x \in \mathcal{X}} \mathbb{E}\left[\mathbb{E}\left[\left|\sum_{i=1}^N \delta(x_{t+1}^i = x) - \sum_{i=1}^N P(x_t^i, u_t^i, \boldsymbol{\mu}_t^N, \boldsymbol{\nu}_t^N)(x)\right|\Big|\boldsymbol{x}_t^N, \boldsymbol{u}_t^N\right]\right]$$

$$\overset{(a)}{\leq} \frac{1}{\sqrt{N}}\sqrt{|\mathcal{X}|}$$

Inequality $(a)$ is obtained applying Lemma 12, and the facts that $\{x_{t+1}^i\}_{i \in \{1,\cdots,N\}}$ are conditionally independent given $\{\boldsymbol{x}_t^N, \boldsymbol{u}_t^N\}$, and the following relations hold

$$\mathbb{E}\left[\delta(x_{t+1}^i = x)\Big|\boldsymbol{x}_t^N, \boldsymbol{u}_t^N\right] = P(x_t^i, u_t^i, \boldsymbol{\mu}_t^N, \boldsymbol{\nu}_t^N)(x),$$

$$\sum_{x \in \mathcal{X}} \mathbb{E}\left[\delta(x_{t+1}^i = x)\Big|\boldsymbol{x}_t^N, \boldsymbol{u}_t^N\right] = 1$$

$\forall i \in \{1,\cdots,N\}$, and $\forall x \in \mathcal{X}$. The second term satisfies the following bound.

$$J_2 \triangleq \frac{1}{N}\sum_{x \in \mathcal{X}} \mathbb{E}\left|\sum_{i=1}^N P(x_t^i, u_t^i, \boldsymbol{\mu}_t^N, \boldsymbol{\nu}_t^N)(x) - \sum_{i=1}^N P(x_t^i, u_t^i, \boldsymbol{\mu}_t^N, \nu^{\mathrm{MF}}(\boldsymbol{\mu}_t^N, \pi_t))(x)\right|$$

$$\leq \frac{1}{N}\sum_{i=1}^N \mathbb{E}\left|P(x_t^i, u_t^i, \boldsymbol{\mu}_t^N, \boldsymbol{\nu}_t^N) - P(x_t^i, u_t^i, \boldsymbol{\mu}_t^N, \nu^{\mathrm{MF}}(\boldsymbol{\mu}_t^N, \pi_t))\right|_1$$

$$\overset{(a)}{\leq} L_P \mathbb{E}\left|\boldsymbol{\nu}_t^N - \nu^{\mathrm{MF}}(\boldsymbol{\mu}_t^N, \pi_t)\right|_1 \overset{(b)}{\leq} \frac{L_P}{\sqrt{N}}\sqrt{|\mathcal{U}|}$$

Inequality (a) is a consequence of Assumption 2 while (b) follows from Lemma 5. Finally, the term, $J_3$ can be upper bounded as follows.

$$J_3 \triangleq \frac{1}{N} \sum_{x \in \mathcal{X}} \mathbb{E} \left| \sum_{i=1}^{N} P(x_t^i, u_t^i, \boldsymbol{\mu}_t^N, \nu^{\mathrm{MF}}(\boldsymbol{\mu}_t^N, \pi_t))(x) - \sum_{i=1}^{N} \sum_{u \in \mathcal{U}} P(x_t^i, u, \boldsymbol{\mu}_t^N, \nu^{\mathrm{MF}}(\boldsymbol{\mu}_t^N, \pi_t))(x) \pi_t(x_t^i, \boldsymbol{\mu}_t^N)(u) \right|$$

$$\overset{(a)}{\leq} \frac{1}{\sqrt{N}} \sqrt{|\mathcal{X}|}$$

Inequality (a) is a result of Lemma 12. In particular, it uses the facts that, $\{u_t^i\}_{i \in \{1, \cdots, N\}}$ are conditionally independent given $\boldsymbol{x}_t^N$, and the following relations hold

$$\mathbb{E} \left[ P(x_t^i, u_t^i, \boldsymbol{\mu}_t^N, \nu^{\mathrm{MF}}(\boldsymbol{\mu}_t^N, \pi_t))(x) \Big| \boldsymbol{x}_t^N \right] = \sum_{u \in \mathcal{U}} P(x_t^i, u, \boldsymbol{\mu}_t^N, \nu^{\mathrm{MF}}(\boldsymbol{\mu}_t^N, \pi_t))(x) \pi_t(x_t^i, \boldsymbol{\mu}_t^N)(u),$$

$$\sum_{x \in \mathcal{X}} \mathbb{E} \left[ P(x_t^i, u_t^i, \boldsymbol{\mu}_t^N, \nu^{\mathrm{MF}}(\boldsymbol{\mu}_t^N, \pi_t))(x) \Big| \boldsymbol{x}_t^N \right] = 1$$

$\forall i \in \{1, \cdots, N\}$, and $\forall x \in \mathcal{X}$. This concludes the Lemma.

## H   Proof of Lemma 7

Observe the following decomposition.

$$\mathbb{E} \left| \frac{1}{N} \sum_{i=1}^{N} r(x_t^i, u_t^i, \boldsymbol{\mu}_t^N, \boldsymbol{\nu}_t^N) - r^{\mathrm{MF}}(\boldsymbol{\mu}_t^N, \pi_t) \right|$$

$$\overset{(a)}{=} \mathbb{E} \left| \frac{1}{N} \sum_{i=1}^{N} r(x_t^i, u_t^i, \boldsymbol{\mu}_t^N, \boldsymbol{\nu}_t^N) - \sum_{x \in \mathcal{X}} \sum_{u \in \mathcal{U}} r(x, u, \boldsymbol{\mu}_t^N, \nu^{\mathrm{MF}}(\boldsymbol{\mu}_t^N, \pi_t)) \pi_t(x, \boldsymbol{\mu}_t^N)(u) \frac{1}{N} \sum_{i=1}^{N} \delta(x_t^i = x) \right|$$

$$= \mathbb{E} \left| \frac{1}{N} \sum_{i=1}^{N} r(x_t^i, u_t^i, \boldsymbol{\mu}_t^N, \boldsymbol{\nu}_t^N) - \frac{1}{N} \sum_{i=1}^{N} \sum_{u \in \mathcal{U}} r(x_t^i, u, \boldsymbol{\mu}_t^N, \nu^{\mathrm{MF}}(\boldsymbol{\mu}_t^N, \pi_t)) \pi_t(x_t^i, \boldsymbol{\mu}_t^N)(u) \right|$$

$$\leq J_1 + J_2$$

Equation (a) uses the definition of $r^{\mathrm{MF}}(\cdot, \cdot)$ as depicted in (9). The term, $J_1$, obeys the following bound.

$$J_1 \triangleq \frac{1}{N} \mathbb{E} \left| \sum_{i=1}^{N} r(x_t^i, u_t^i, \boldsymbol{\mu}_t^N, \boldsymbol{\nu}_t^N) - \sum_{i=1}^{N} r(x_t^i, u_t^i, \boldsymbol{\mu}_t^N, \nu^{\mathrm{MF}}(\boldsymbol{\mu}_t^N, \pi_t)) \right|$$

$$\leq \frac{1}{N} \mathbb{E} \sum_{i=1}^{N} \left| r(x_t^i, u_t^i, \boldsymbol{\mu}_t^N, \boldsymbol{\nu}_t^N) - r(x_t^i, u_t^i, \boldsymbol{\mu}_t^N, \nu^{\mathrm{MF}}(\boldsymbol{\mu}_t^N, \pi_t)) \right|$$

$$\overset{(a)}{\leq} L_R \mathbb{E} \left| \boldsymbol{\nu}_t^N - \nu^{\mathrm{MF}}(\boldsymbol{\mu}_t^N, \pi_t) \right|_1$$

$$\overset{(b)}{\leq} \frac{L_R}{\sqrt{N}} \sqrt{|\mathcal{U}|}$$

Inequality (a) results from Assumption 1, whereas (b) is a consequence of Lemma 5. The term, $J_2$, satisfies the following.

$$J_2 \triangleq \frac{1}{N}\mathbb{E}\left|\sum_{i=1}^{N} r(x_t^i, u_t^i, \boldsymbol{\mu}_t^N, \nu^{\mathrm{MF}}(\boldsymbol{\mu}_t^N, \pi_t)) - \sum_{i=1}^{N}\sum_{u\in\mathcal{U}} r(x_t^i, u, \boldsymbol{\mu}_t^N, \nu^{\mathrm{MF}}(\boldsymbol{\mu}_t^N, \pi_t))\pi_t(x_t^i, \boldsymbol{\mu}_t^N)(u)\right|$$

$$= \frac{1}{N}\mathbb{E}\left[\mathbb{E}\left[\left|\sum_{i=1}^{N} r(x_t^i, u_t^i, \boldsymbol{\mu}_t^N, \nu^{\mathrm{MF}}(\boldsymbol{\mu}_t^N, \pi_t)) - \sum_{i=1}^{N}\sum_{u\in\mathcal{U}} r(x_t^i, u, \boldsymbol{\mu}_t^N, \nu^{\mathrm{MF}}(\boldsymbol{\mu}_t^N, \pi_t))\pi_t(x_t^i, \boldsymbol{\mu}_t^N)(u)\right|\,\Big|\,\boldsymbol{x}_t^N\right]\right]$$

$$= \frac{M_R}{N}\mathbb{E}\left[\mathbb{E}\left[\left|\sum_{i=1}^{N} r_0(x_t^i, u_t^i, \boldsymbol{\mu}_t^N, \nu^{\mathrm{MF}}(\boldsymbol{\mu}_t^N, \pi_t)) - \sum_{i=1}^{N}\sum_{u\in\mathcal{U}} r_0(x_t^i, u, \boldsymbol{\mu}_t^N, \nu^{\mathrm{MF}}(\boldsymbol{\mu}_t^N, \pi_t))\pi_t(x_t^i, \boldsymbol{\mu}_t^N)(u)\right|\,\Big|\,\boldsymbol{x}_t^N\right]\right]$$

$$\overset{(a)}{\leq} \frac{M_R}{\sqrt{N}}$$

where $r_0(\cdot,\cdot,\cdot,\cdot) \triangleq r(\cdot,\cdot,\cdot,\cdot)/M_R$. Inequality (a) follows from Lemma 12. In particular, it utilises the fact that $\{u_t^i\}_{i\in\{1,\cdots,N\}}$ are conditionally independent given $\boldsymbol{x}_t$, and the following relations hold.

$$|r_0(x_t^i, u_t^i, \boldsymbol{\mu}_t^N, \nu^{\mathrm{MF}}(\boldsymbol{\mu}_t^N, \pi_t))| \leq 1,$$

$$\mathbb{E}\left[r_0(x_t^i, u_t^i, \boldsymbol{\mu}_t^N, \nu^{\mathrm{MF}}(\boldsymbol{\mu}_t^N, \pi_t))\Big|\boldsymbol{x}_t^N\right] = \sum_{u\in\mathcal{U}} r_0(x_t^i, u, \boldsymbol{\mu}_t^N, \nu^{\mathrm{MF}}(\boldsymbol{\mu}_t^N, \pi_t))\pi_t(x_t^i, \boldsymbol{\mu}_t^N)(u)$$

$\forall i \in \{1, \cdots, N\}, \forall u \in \mathcal{U}$.

# I  Proof of Lemma 8

Observe that,

$$\mathbb{E}|\boldsymbol{\mu}_t^N - \boldsymbol{\mu}_t^\infty|_1 \leq \mathbb{E}\left|\boldsymbol{\mu}_t^N - P^{\mathrm{MF}}(\boldsymbol{\mu}_{t-1}^N, \pi_{t-1})\right|_1 + \mathbb{E}\left|P^{\mathrm{MF}}(\boldsymbol{\mu}_{t-1}^N, \pi_{t-1}) - \boldsymbol{\mu}_t^\infty\right|_1$$

$$\overset{(a)}{\leq} \frac{C_P}{\sqrt{N}}\left[\sqrt{|\mathcal{X}|} + \sqrt{|\mathcal{U}|}\right] + \mathbb{E}\left|P^{\mathrm{MF}}(\boldsymbol{\mu}_{t-1}^N, \pi_{t-1}) - P^{\mathrm{MF}}(\boldsymbol{\mu}_{t-1}^\infty, \pi_{t-1})\right|_1$$

Inequality (a) follows from Lemma 6, and relation (8). Using Lemma 3, we get

$$\left|P^{\mathrm{MF}}(\boldsymbol{\mu}_{t-1}^N, \pi_{t-1}) - P^{\mathrm{MF}}(\boldsymbol{\mu}_{t-1}^\infty, \pi_{t-1})\right|_1$$

$$\leq \tilde{S}_P|\boldsymbol{\mu}_{t-1}^N - \boldsymbol{\mu}_{t-1}^\infty|_1 + \bar{S}_P|\pi_{t-1}(\cdot, \boldsymbol{\mu}_{t-1}^N) - \pi_{t-1}(\cdot, \boldsymbol{\mu}_{t-1}^\infty)|_\infty$$

$$\overset{(a)}{\leq} S_P|\boldsymbol{\mu}_{t-1}^N - \boldsymbol{\mu}_{t-1}^\infty|_1$$

where $S_P \triangleq \tilde{S}_P + L_Q\bar{S}_P$. Inequality (a) follows from Assumption 3. Combining, we get,

$$\mathbb{E}|\boldsymbol{\mu}_t^N - \boldsymbol{\mu}_t^\infty|_1 \leq \frac{C_P}{\sqrt{N}}\left[\sqrt{|\mathcal{X}|} + \sqrt{|\mathcal{U}|}\right] + S_P\mathbb{E}\left|\boldsymbol{\mu}_{t-1}^N - \boldsymbol{\mu}_{t-1}^\infty\right|_1 \tag{44}$$

Recursively applying the above inequality, we finally obtain,

$$\mathbb{E}|\boldsymbol{\mu}_t^N - \boldsymbol{\mu}_t^\infty|_1 \leq \frac{C_P}{\sqrt{N}}\left[\sqrt{|\mathcal{X}|} + \sqrt{|\mathcal{U}|}\right]\left(\frac{S_P^t - 1}{S_P - 1}\right)$$

## J  Proof of Lemma 9

Note that,

$$
\begin{aligned}
&\mathbb{E}\left|\boldsymbol{\mu}_{t+1}^N - P^{\mathrm{MF}}(\boldsymbol{\mu}_t^N, \pi_t)\right|_1 \\
&\overset{(a)}{=} \sum_{x \in \mathcal{X}} \mathbb{E}\left|\frac{1}{N}\sum_{i=1}^N \delta(x_{t+1}^i = x) - \sum_{x' \in \mathcal{X}}\sum_{u \in \mathcal{U}} P(x', u, \boldsymbol{\mu}_t^N)(x)\pi_t(x', \boldsymbol{\mu}_t^N)(u)\frac{1}{N}\sum_{i=1}^N \delta(x_t^i = x')\right| \\
&= \sum_{x \in \mathcal{X}} \mathbb{E}\left|\frac{1}{N}\sum_{i=1}^N \delta(x_{t+1}^i = x) - \frac{1}{N}\sum_{i=1}^N\sum_{u \in \mathcal{U}} P(x_t^i, u, \boldsymbol{\mu}_t^N)(x)\pi_t(x_t^i, \boldsymbol{\mu}_t^N)(u)\right| \\
&\leq J_1 + J_2
\end{aligned}
$$

Equality (a) follows from the definition of $P^{\mathrm{MF}}(\cdot, \cdot)$ as depicted in (8). The first term, $J_1$, can be upper bounded as follows.

$$
\begin{aligned}
J_1 &\triangleq \frac{1}{N}\sum_{x \in \mathcal{X}} \mathbb{E}\left|\sum_{i=1}^N \delta(x_{t+1}^i = x) - \sum_{i=1}^N P(x_t^i, u_t^i, \boldsymbol{\mu}_t^N)(x)\right| \\
&= \frac{1}{N}\sum_{x \in \mathcal{X}} \mathbb{E}\left[\mathbb{E}\left[\left|\sum_{i=1}^N \delta(x_{t+1}^i = x) - \sum_{i=1}^N P(x_t^i, u_t^i, \boldsymbol{\mu}_t^N)(x)\right|\,\bigg|\,\boldsymbol{x}_t^N, \boldsymbol{u}_t^N\right]\right] \\
&\overset{(a)}{\leq} \frac{1}{\sqrt{N}}\sqrt{|\mathcal{X}|}
\end{aligned}
$$

Inequality (a) can be derived using Lemma 12, and the facts that $\{x_{t+1}^i\}_{i \in \{1, \cdots, N\}}$ are conditionally independent given $\{\boldsymbol{x}_t^N, \boldsymbol{u}_t^N\}$, and,

$$
\begin{aligned}
\mathbb{E}\left[\delta(x_{t+1}^i = x)\,\big|\,\boldsymbol{x}_t^N, \boldsymbol{u}_t^N\right] &= P(x_t^i, u_t^i, \boldsymbol{\mu}_t^N)(x), \\
\sum_{x \in \mathcal{X}} \mathbb{E}\left[\delta(x_{t+1}^i = x)\,\big|\,\boldsymbol{x}_t^N, \boldsymbol{u}_t^N\right] &= 1
\end{aligned}
$$

$\forall i \in \{1, \cdots, N\}$, and $\forall x \in \mathcal{X}$. The second term can be bounded as follows.

$$
\begin{aligned}
J_2 &\triangleq \frac{1}{N}\sum_{x \in \mathcal{X}} \mathbb{E}\left|\sum_{i=1}^N P(x_t^i, u_t^i, \boldsymbol{\mu}_t^N)(x) - \sum_{i=1}^N\sum_{u \in \mathcal{U}} P(x_t^i, u, \boldsymbol{\mu}_t^N)(x)\pi_t(x_t^i, \boldsymbol{\mu}_t^N)(u)\right| \\
&\overset{(a)}{\leq} \frac{1}{\sqrt{N}}\sqrt{|\mathcal{X}|}
\end{aligned}
$$

Inequality (a) is a consequence of Lemma 12. Specifically, it uses the facts that, $\{u_t^i\}_{i \in \{1, \cdots, N\}}$ are conditionally independent given $\boldsymbol{x}_t^N$, and

$$
\begin{aligned}
\mathbb{E}\left[P(x_t^i, u_t^i, \boldsymbol{\mu}_t^N)(x)\,\big|\,\boldsymbol{x}_t^N\right] &= \sum_{u \in \mathcal{U}} P(x_t^i, u, \boldsymbol{\mu}_t^N)(x)\pi_t(x_t^i, \boldsymbol{\mu}_t^N)(u), \\
\sum_{x \in \mathcal{X}} \mathbb{E}\left[P(x_t^i, u_t^i, \boldsymbol{\mu}_t^N)(x)\,\big|\,\boldsymbol{x}_t^N\right] &= 1
\end{aligned}
$$

$\forall i \in \{1, \cdots, N\}$, and $\forall x \in \mathcal{X}$. This concludes the Lemma.

## K    Proof of Lemma 10

Note that,

$$
\mathbb{E}\left|\frac{1}{N}\sum_{i=1}^{N} r(x_t^i, u_t^i, \boldsymbol{\mu}_t^N) - r^{\mathrm{MF}}(\boldsymbol{\mu}_t^N, \pi_t)\right|
$$

$$
\overset{(a)}{=} \mathbb{E}\left|\frac{1}{N}\sum_{i=1}^{N} r(x_t^i, u_t^i, \boldsymbol{\mu}_t^N) - \sum_{x\in\mathcal{X}}\sum_{u\in\mathcal{U}} r(x, u, \boldsymbol{\mu}_t^N)\pi_t(x, \boldsymbol{\mu}_t^N)(u)\frac{1}{N}\sum_{i=1}^{N}\delta(x_t^i = x)\right|
$$

$$
= \mathbb{E}\left|\frac{1}{N}\sum_{i=1}^{N} r(x_t^i, u_t^i, \boldsymbol{\mu}_t^N) - \frac{1}{N}\sum_{i=1}^{N}\sum_{u\in\mathcal{U}} r(x_t^i, u, \boldsymbol{\mu}_t^N)\pi_t(x_t^i, \boldsymbol{\mu}_t^N)(u)\right|
$$

$$
= \frac{1}{N}\mathbb{E}\left[\mathbb{E}\left[\left|\sum_{i=1}^{N} r(x_t^i, u_t^i, \boldsymbol{\mu}_t^N) - \sum_{i=1}^{N}\sum_{u\in\mathcal{U}} r(x_t^i, u, \boldsymbol{\mu}_t^N)\pi_t(x_t^i, \boldsymbol{\mu}_t^N)(u)\right|\,\middle|\,\boldsymbol{x}_t^N\right]\right]
$$

$$
= \frac{M_R}{N}\mathbb{E}\left[\mathbb{E}\left[\left|\sum_{i=1}^{N} r_0(x_t^i, u_t^i, \boldsymbol{\mu}_t^N) - \sum_{i=1}^{N}\sum_{u\in\mathcal{U}} r_0(x_t^i, u, \boldsymbol{\mu}_t^N)\pi_t(x_t^i, \boldsymbol{\mu}_t^N)(u)\right|\,\middle|\,\boldsymbol{x}_t^N\right]\right]
$$

$$
\overset{(a)}{\leq} \frac{M_R}{\sqrt{N}}
$$

where $r_0(\cdot, \cdot, \cdot, \cdot) \triangleq r(\cdot, \cdot, \cdot, \cdot)/M_R$. Inequality (a) follows from Lemma 12. Specifically, it uses the fact that $\{u_t^i\}_{i\in\{1,\cdots,N\}}$ are conditionally independent given $\boldsymbol{x}_t^N$, and

$$
|r_0(x_t^i, u_t^i, \boldsymbol{\mu}_t^N)| \leq 1,
$$
$$
\mathbb{E}\left[r_0(x_t^i, u_t^i, \boldsymbol{\mu}_t^N)\middle|\boldsymbol{x}_t^N\right] = \sum_{u\in\mathcal{U}} r_0(x_t^i, u, \boldsymbol{\mu}_t^N)\pi_t(x_t^i, \boldsymbol{\mu}_t^N)(u)
$$

$\forall i \in \{1, \cdots, N\}, \forall u \in \mathcal{U}$.

## L    Proof of Lemma 11

Observe that,

$$
\mathbb{E}|\boldsymbol{\mu}_t^N - \boldsymbol{\mu}_t^\infty|_1 \leq \mathbb{E}\left|\boldsymbol{\mu}_t^N - P^{\mathrm{MF}}(\boldsymbol{\mu}_{t-1}^N, \pi_{t-1})\right|_1 + \mathbb{E}\left|P^{\mathrm{MF}}(\boldsymbol{\mu}_{t-1}^N, \pi_{t-1}) - \boldsymbol{\mu}_t^\infty\right|_1
$$
$$
\overset{(a)}{\leq} \frac{2}{\sqrt{N}}\sqrt{|\mathcal{X}|} + \mathbb{E}\left|P^{\mathrm{MF}}(\boldsymbol{\mu}_{t-1}^N, \pi_{t-1}) - P^{\mathrm{MF}}(\boldsymbol{\mu}_{t-1}^\infty, \pi_{t-1})\right|_1
$$

Inequality (a) follows from Lemma 6, and relation (8). Using Lemma 3, we get

$$
\left|P^{\mathrm{MF}}(\boldsymbol{\mu}_{t-1}^N, \pi_{t-1}) - P^{\mathrm{MF}}(\boldsymbol{\mu}_{t-1}^\infty, \pi_{t-1})\right|_1
$$
$$
\leq \tilde{S}_P|\boldsymbol{\mu}_{t-1}^N - \boldsymbol{\mu}_{t-1}^\infty|_1 + \bar{S}_P|\pi_{t-1}(\cdot, \boldsymbol{\mu}_{t-1}^N) - \pi_{t-1}(\cdot, \boldsymbol{\mu}_{t-1}^\infty)|_\infty
$$
$$
\overset{(a)}{\leq} S_P|\boldsymbol{\mu}_{t-1}^N - \boldsymbol{\mu}_{t-1}^\infty|_1
$$

where $S_P \triangleq \tilde{S}_P + L_Q\bar{S}_P$. Inequality (a) follows from Assumption 3. Combining, we get,

$$
\mathbb{E}|\boldsymbol{\mu}_t^N - \boldsymbol{\mu}_t^\infty|_1 \leq \frac{2}{\sqrt{N}}\sqrt{|\mathcal{X}|} + S_P\mathbb{E}\left|\boldsymbol{\mu}_{t-1}^N - \boldsymbol{\mu}_{t-1}^\infty\right|_1 \tag{45}
$$

Recursively applying the above inequality, we finally obtain,

$$
\mathbb{E}|\boldsymbol{\mu}_t^N - \boldsymbol{\mu}_t^\infty|_1 \leq \frac{2}{\sqrt{N}}\sqrt{|\mathcal{X}|}\left(\frac{S_P^t - 1}{S_P - 1}\right)
$$

This concludes the Lemma.

# M Sampling Algorithm

---

**Algorithm 3** Sampling Algorithm

---

**Input:** $\boldsymbol{\mu}_0$, $\boldsymbol{\pi}_{\Phi_j}$, $P$, $r$

1: Sample $x_0 \sim \boldsymbol{\mu}_0$.
2: Sample $u_0 \sim \pi_{\Phi_j}(x_0, \boldsymbol{\mu}_0)$
3: $\boldsymbol{\nu}_0 \leftarrow \nu^{\mathrm{MF}}(\boldsymbol{\mu}_0, \pi_{\Phi_j})$

4: $t \leftarrow 0$
5: FLAG $\leftarrow$ FALSE
6: **while** FLAG is FALSE **do**
7: $\quad$ FLAG $\leftarrow$ TRUE with probability $1 - \gamma$.
8: $\quad$ Execute Update
9: **end while**
10: $T \leftarrow t$
11: Accept $(x_T, \boldsymbol{\mu}_T, u_T)$ as a sample.

12: $\hat{V}_{\Phi_j} \leftarrow 0$, $\hat{Q}_{\Phi_j} \leftarrow 0$
13: FLAG $\leftarrow$ FALSE
14: SumRewards $\leftarrow 0$
15: **while** FLAG is FALSE **do**
16: $\quad$ FLAG $\leftarrow$ TRUE with probability $1 - \gamma$.
17: $\quad$ Execute Update
18: $\quad$ SumRewards $\leftarrow$ SumRewards $+ r(x_t, u_t, \boldsymbol{\mu}_t, \boldsymbol{\nu}_t)$
19: **end while**

20: With probability $\frac{1}{2}$, $\hat{V}_{\Phi_j} \leftarrow$ SumRewards. Otherwise $\hat{Q}_{\Phi_j} \leftarrow$ SumRewards.
21: $\hat{A}_{\Phi_j}(x_T, \boldsymbol{\mu}_T, u_T) \leftarrow 2(\hat{Q}_{\Phi_j} - \hat{V}_{\Phi_j})$.

**Output:** $(x_T, \boldsymbol{\mu}_T, u_T)$ and $\hat{A}_{\Phi_j}(x_T, \boldsymbol{\mu}_T, u_T)$

**Procedure** Update:

1: $x_{t+1} \sim P(x_t, u_t, \boldsymbol{\mu}_t, \boldsymbol{\nu}_t)$.
2: $\boldsymbol{\mu}_{t+1} \leftarrow P^{\mathrm{MF}}(\boldsymbol{\mu}_t, \pi_{\Phi_j})$
3: $u_{t+1} \sim \pi_{\Phi_j}(x_{t+1}, \boldsymbol{\mu}_{t+1})$
4: $\boldsymbol{\nu}_{t+1} \leftarrow \nu^{\mathrm{MF}}(\boldsymbol{\mu}_{t+1}, \pi_{\Phi_j})$
5: $t \leftarrow t + 1$

**EndProcedure**

---

# N   Proof of Theorem 3

Note that,

$$\left| \sup_{\Phi \in \mathbb{R}^d} v_{\text{MARL}}(\boldsymbol{\mu}_0, \pi_\Phi) - \frac{1}{J} \sum_{j=1}^J v_{\text{MARL}}(\boldsymbol{\mu}_0, \tilde{\pi}_{\Phi_j}) \right|$$

$$\leq \left| \sup_{\Phi \in \mathbb{R}^d} v_{\text{MARL}}(\boldsymbol{\mu}_0, \pi_\Phi) - \sup_{\Phi \in \mathbb{R}^d} v_{\text{MF}}(\boldsymbol{\mu}_0, \pi_\Phi) \right| + \underbrace{\left| \sup_{\Phi \in \mathbb{R}^d} v_{\text{MF}}(\boldsymbol{\mu}_0, \pi_\Phi) - \frac{1}{J} \sum_{j=1}^J v_{\text{MF}}(\boldsymbol{\mu}_0, \pi_{\Phi_j}) \right|}_{\triangleq J_2}$$

$$+ \underbrace{\frac{1}{J} \sum_{j=1}^J \left| v_{\text{MF}}(\boldsymbol{\mu}_0, \pi_{\Phi_j}) - v_{\text{MARL}}(\boldsymbol{\mu}_0, \tilde{\pi}_{\Phi_j}) \right|}_{\triangleq J_3}$$

$$\leq \underbrace{\sup_{\Phi \in \mathbb{R}^d} \left| v_{\text{MARL}}(\boldsymbol{\mu}_0, \pi_\Phi) - v_{\text{MF}}(\boldsymbol{\mu}_0, \pi_\Phi) \right| + J_2 + J_3}_{\triangleq J_1}$$

Using the same argument as used in Appendix A.3, we can conclude that,

$$J_1 \leq \frac{C_1}{\sqrt{N}} \left[ \sqrt{|\mathcal{X}|} + \sqrt{|\mathcal{U}|} \right] = C_1 e \leq C_1 \max\{e, \epsilon\},$$

$$J_3 \leq \frac{C_3}{\sqrt{N}} \left[ \sqrt{|\mathcal{X}|} + \sqrt{|\mathcal{U}|} \right] = C_3 e \leq C_2 \max\{e, \epsilon\}$$

for some constants $C_1, C_3$. Moreover, Lemma 1 suggests that,

$$J_2 \leq \frac{\sqrt{\epsilon_{\text{bias}}}}{1 - \gamma} + \epsilon \leq \frac{\sqrt{\epsilon_{\text{bias}}}}{1 - \gamma} + \max\{e, \epsilon\}$$

Taking $C = C_1 + C_3 + 1$, we conclude the result. Under Assumption 4, using the same argument as is used in Appendix B.2, we can improve the bounds on $J_1, J_3$ as follows.

$$J_1 \leq \frac{C_1}{\sqrt{N}} \sqrt{|\mathcal{X}|},$$

$$J_3 \leq \frac{C_3}{\sqrt{N}} \sqrt{|\mathcal{X}|}$$

This establishes the improved result.

