# OpenReview forum: "On the Near-Optimality of Local Policies in Large Cooperative Multi-Agent Reinforcement Learning"
_TMLR — Accepted by TMLR_

### Review · Reviewer_1L6s · 2022-07-17

**Summary Of Contributions:**

This paper studied the cooperate N-agent network problem in which the agents cooperate to achieve the optimal $\pi^*_{MARL}$. Previous works require global execution in order to achieve the theoretical guarantee. However, in many applications, global information is impossible to collect and thus it is desirable to have a locally executable policy such as it still has theoretical guarantee. The authors suggest such algorithm which an approximation error of $O(\frac{1}{\sqrt{N}}(\sqrt{|X|}+\sqrt{|U|}))$.

**Requested Changes:**

There is no major changes needed to make for now. I may need to add more during the discussion phase.

**Strengths And Weaknesses:**

I believe the direction that the paper takes is very important. The paper is well written and easy to follow. To the best of my knowledge, all the proves are rigorous. The observation that each agent can approximate the state distribution of a finite agent system by the infinite agent system in MFC (which can be obtained locally) is intuitive and new.

The work of Gu et al., (2021) shows the approximation of $O(\frac{1}{\sqrt{N}})$ between MF strategy and the optimal MARL strategy. In this paper, by replacing $\mu_t^N$ with $\mu_t^{\infty}$, it can overcome the costly procedure of computing empirical state distribution. Therefore, following the work of Gu et al., (2021), theoretically, this paper only needs to bound the performance of $\tilde{\pi}^*_{MF}$ and ${\pi}^*_{MF}$. However, in order to achieve the local execution, the current approximation error depends on the number of state and action $|X|$ and $|U|$, which will limit the application of the algorithm in many cases where the size of state or action are large. The authors find a way to solve this problem in special case where the reward function and transition function are independent of the action distribution of the population. In this case, the approximation error only depends on the size of state space $|X|$.

In the experiment section, can the authors provide results when both size of state and action space are large so that we can observe how severely the approximation will be affected?


Minors: Equation (2): $(x)$ should be $(u)$.

---

> ### Author Response · Authors · 2022-08-02
> **Response to Reviewer 1L6s**
>
> $\textbf{Q1: Result demonstrating how severely}~|\mathcal{X}| ~\textbf{impacts the approximation error:}$
>
> In the revised manuscript, we have added the requested result (Fig. 1b). The figure can also be accessed via the following anonymous link: https://anonymous.4open.science/r/LocalPolicy-A4FE/Display/Fig1b.png
>
> $\textbf{Q2: Minor comment:}$
>
> Thank you for pointing out the typo. We have corrected the mistake in the revised manuscript.

---

### Review · Reviewer_HZgm · 2022-07-23

**Summary Of Contributions:**

The paper aims to design locally executable policies with optimality guarantees.
1.	The authors demonstrate that given an initial state distribution, it is possible to obtain a non-stationary locally executable policy-sequence $\tilde{\pi}$ that the average time-discounted sum of rewards generated by $\tilde{\pi}$ closely approximates the value generated by the optimal policy sequence $\pi^*$.
2.	The authors use the idea of mean field control to solve the scalability problem in large-scale network. The authors show that, under the conditions that several common assumptions hold, the approximation error between the optimal control policy obtained using mean-field control and the real optimal control policy in MARL is tolerable, which is given by $O(\frac{1}{\sqrt{N}}) [\sqrt{|\mathcal{X}|} + \sqrt{|\mathcal{U}|}]$. Besides, they also prove that if the reward and transition functions do not depends on the action distribution of the population, the optimality gap for local policies can be bounded as $O(\frac{1}{\sqrt{N}}) [\sqrt{|\mathcal{X}|} ]$.
3.	The authors also discussed how near-optimal polices can be obtained. Firstly, a NPG (natural policy gradient) based algorithm is used to approximately obtain $\pi_{MF}^*$, then the obtained policy will be localized and executed in a decentralized manner.  The paper also demonstrates a numerical experiment to support the theoretical results.

**Broader Impact Concerns:**

The paper illustrates the optimality gap between the optimal control policy obtained using mean-field control and the real optimal control policy in a MARL setting and presents a practical NPG-based algorithm to search for it in a scalable manner.  The case where the interactions between the agents are non-uniform is worth explored as the authors state in Section 9. Besides, the framework can also be exploited in the setting of other problem, such as LQR control. And other gradient methods (like PG case or Gauss-Newton case) can also be considered. This paper also provides theoretical support for potential practical applications.

**Requested Changes:**

1.	Page 7, the equation between equation (16) and (17), it would be better to add absolute value notation. Although the result is correct, adding the absolute value symbol will make it easier for the readers to understand.
2.	In the numerical experiment section, the authors are supposed to clarify whether all the assumptions made in the context are satisfied in the setting.
3.	It would be better to give some concrete examples that satisfy the assumption 4 in the special case (page 5).

**Strengths And Weaknesses:**

Strengths:
1.	The paper innovatively tries to solve the policy optimization problem in MARL under the mean-field control framework. To my best knowledge, there has been some work in the past that exploits the spatial structure of networks to address the curse of dimensionality in large-scale networks control. It provides a new way of thinking to analyze the problem from the perspective of mean-field control.
2.	The article's analysis of the optimality gap is intuitive, and the proof process is solid as well.
3.	The author gives a very detailed discussion in terms of optimality theory, algorithm and even some special cases. The logic of the article is strict and exhaustive.
Weaknesses:
1.	Interpretation could be deepened regarding the numerical experiment.  There are many assumptions made in the paper, and it would be better for authors to discuss in detail whether these assumptions hold in the current setting of the numerical experiment.
2.	For the additional assumption (assumption 4) in the special case, the author is supposed to give some concrete examples to illustrate their rationality.

---

> ### Author Response · Authors · 2022-08-02
> **Response to Reviewer HZgm**
>
> $\textbf{Q1: On the equation between (16) and (17):}$
>
> In the equation mentioned above, we bounded the term $v\_{\mathrm{MARL}}(\boldsymbol{x}\_0\^N, \boldsymbol{\pi}\^\*\_{\mathrm{MARL}})$ - $v\_{\mathrm{MARL}}(\boldsymbol{x}\_0\^N, \boldsymbol{\tilde{\pi}}\^\*\_{\mathrm{MF}})$. In the paragraph following this equation, we mentioned that one can similarly bound its negation $v\_{\mathrm{MARL}}(\boldsymbol{x}\_0\^N, \boldsymbol{\tilde{\pi}}\^\*\_{\mathrm{MF}})-v\_{\mathrm{MARL}}(\boldsymbol{x}\_0\^N, \boldsymbol{\pi}\^\*\_{\mathrm{MARL}})$. Combining these two, one can therefore bound $|v\_{\mathrm{MARL}}(\boldsymbol{x}\_0\^N, \boldsymbol{\pi}\^\*\_{\mathrm{MARL}})-v\_{\mathrm{MARL}}(\boldsymbol{x}\_0\^N, \boldsymbol{\tilde{\pi}}\^\*\_{\mathrm{MF}})|$. We have elaborated this argument in the revised manuscript.
>
> $\textbf{Q2: Are the assumptions satisfied?}$
>
> We have clarified in Section 8 of the revised manuscript that the experimental setup considered in our paper satisfies assumptions $1-4$.
>
> $\textbf{Q3: Concrete examples satisfying assumption 4:}$
>
> We have mentioned in Section 8 of the revised manuscript that the experimental setup considered in our paper satisfies assumption 4.

---

### Review · Reviewer_84q8 · 2022-07-25

**Summary Of Contributions:**

This paper considers a multi-agent optimization problem in which the interactions between agents of of mean field type (this means that that the rewards and behavior of one agent depend on other agents only through their empirical distribution of states and actions). A controller wants to optimize the expected discounted reward over all agents. The problem is motivated by multi-agent reinforcement learning problems.

The paper contains two parts. The first one concerns mean field control. The authors show that there exists a (time-dependent) decentralized controller that attains a performance that is at most C/\sqrt(N) of the best centralized controller.  An upper bound on the constant C is computed (note that it is likely to be infinite for large discount factors).  The second part concerns learning. The authors show how a natural policy gradient algorithm can be implemented in a decentralized way (although I am not sure to have understood if the algorithm is fully decentralized).



**Broader Impact Concerns:**

I do not think that a broader impact statement is needed. The paper is mostly of theoretical interest.

**Requested Changes:**

Correct or discuss the 3 weaknesses mentioned above:
- Better discuss the relationship with the existing literature on mean field control and the bounds that already exist.
- Explicit the dependence of the bounds on the parameters
- Clarify Section 7, notably by using more precise definitions and notations.


**Strengths And Weaknesses:**

Strengths* -- The paper is well written. Although I did not check all details of the proofs, the methodology is sound and I believe that the obtained bounds are correct. I believe that the bounds are new, even if I there are more results in the related work on mean field control than what the authors claim.

*Weakness 1: related work* --  most of the paper is focussed on studying the link between the control of a system with N objects and the mean field control problem and to obtain central-limit-theorem like results. While I believe that the bounds presented in the paper are new, these results are really close to the one provided in Theorem 55 and Proposition 12 of (Gast and Gaujal 2011): the fact that deterministic (a.k.a. time-dependent local policies) are optimal is also present in this paper.  Hence, a better discussion with the literature on mean field control and the bounds obtained in such cases is I believe missing.

*Weakness 2: dependence of the bounds on parameters*
- Also, the authors greatly discuss the dependence in \sqrt{|X|} or \sqrt{|U|} in the bounds. Yet, given that the bounds involve Lipchitz constants, the Lipchitz constant might also hide factors that are multiple of \sqrt{|X|} (e.g. think of L_\infty, L_2 or L_1 norms).  Is What is the impact of using one norm or another one?
- When \gamma is close to 1 the constant is infinity because \gamma S_P will be larger than 1. I believe that this will essentially always be the case because S_P is likely to be at least 3 (unless L_P is smaller than 1). Moreover the dependence on L_Q seems strong and it seems that it is not present in (Gast and Gaujal 2011): is this dependence needed? Or is it a proof artifact? Or did I miss a difference between the results of (Gast and Gaujal 2011) and this paper?

*Weakness 3: learning part* -- I am a bit puzzled by the learning part and I did not understand what is accessible to the learner and what is assumed to be known. I find the whole part quite unclear. In particular:
- is Algorithm 1 specific to the current problem or is it just a reminder of an existing algorithm?
- in Algorithm 2: how does line 3 works? Does each agent need to know the global P?
- Equation below equation (20): either the notation is wrong or this does not defind a distribution. This might be due to multiple use of the variables (x,\mu,u) having different meanings.  Similarly in Equation (19), the "u" seems to be once a deterministic variable, once a random variables in the expression A(u) = Q(u) - E(Q(u)).
- sentences like "The sample complexity of Algorithm 1 to achieve (22) is O(\epsilon^{-3})" are not precise enough for me: what is hidden in the O(.)? Do you consider a generative model?

*Other comments* --  In the conclusion, authors talk about "non-unniform" interactions. There has been some progress on mean field about what is called "heterogeneous" or "non-uniform" models. On the topic of optimal control, one can refer to the paper by Brown and Smith (2020). This paper is about restless bandit, which are not the same model as the mean field control considered in this paper because the evolution of agents and constraints on the actions are not the same but they share the same "mean field interaction" structure.


References:
- (Gast and Gaujal 2011) "A mean field approach for optimization in discrete time" N Gast, B Gaujal - Discrete Event Dynamic Systems, 2011
- (Brown and Smith 2020) Index policies and performance bounds for dynamic selection problems DB Brown, JE Smith - Management Science, 2020

---

> ### Author Response · Authors · 2022-08-02
> **Response to Reviewer 84q8: Part I**
>
> $\textbf{Q1: Comparison with the existing literature:}$
>
>  We would like to point out that the problem setting considered in (Gast and Gaujal 2011) is significantly different from the multi-agent reinforcement learning (MARL) setup considered in our paper. Below we enlist several key differences between these models.
>
> 1. (Gast and Gaujal 2011) considers a system that comprises of a single controller and $N$ objects. The objects are assumed to have no capability to execute actions. Only the controller is endowed with that responsibility. In contrast, our article considers a collection of $N$ interacting agents, each associated with an action set, $\mathcal{U}$.
>
> 2. The $N$ objects described in the cited paper have no associated rewards. Only the controller is endowed with a reward function. In contrast, all agents in our model receive rewards at each time instant.
>
> 3. Moreover, the reward received by the controller in (Gast and Gaujal 2011)  is independent of its actions. On the contrary, the reward of each agent in our model, in general, is a function of the joint action of all the agents.
>
> We would like to further clarify that (Gast and Gaujal 2011) establishes the optimality of a deterministic policy. However, a deterministic policy, in general, can be non-local whereas a local policy, in general, can be non-deterministic. A deterministic policy simply generates a constant (non-probabilistic) action for each of its state inputs. A local policy, on the other hand, might yield actions probabilistically, but only takes local states as an input. Therefore, the locality and stochasticity of a policy are not mutually exclusive properties. In our article, we obtain local policies by replacing a stochastic variable, $\boldsymbol{\mu}_t^N$ ($N$-agent state distribution at time $t$) with a deterministic variable, $\boldsymbol{\mu}_t^\infty$ (mean-field state distribution at time $t$) as an input to the policy function. We hope this clarifies the confusion.
>
> In the revised manuscript (Section 1.2), we have cited (Gast and Gaujal 2011) and compared its $N$-object model with our MARL model. Specifically, the following paragraph has been added.
>
> "Alongside standard MARL problems, the concept of MFC-based solutions has also been applied to other variants of reinforcement learning (RL) problems. For example, (Gast and Gaujal 2011) considers a system comprising of a single controller and $N$ bodies. The controller is solely responsible for taking actions and the bodies are associated with states that change as functions of the chosen action. At each time instant, the controller receives a reward that solely is a function of the current states. The objective is to strategize the choice of actions as a function of states such that the cumulative reward of the controller is maximized. It is proven that the above problem can be approximated via a mean-field process."

---

> ### Author Response · Authors · 2022-08-02
> **Response to Reviewer 84q8: Part II**
>
> $\textbf{Q2: Dependence of Lipschitz Constants on the sizes of the state and action spaces:}$
>
> Lipschitz constants are arbitrary positive reals that, in general, have no connections with $|\mathcal{X}|$, and $|\mathcal{U}|$. Changing the norm will change the approximation error calculations, while the Lipschitz constants would still not depend on the sizes of the state and action spaces.
>
> How the error changes for $L_p$, norm $p\geq 1$  is an interesting question, and here we provide an elementary argument to obtain a preliminary result. Note that the Lipschitz continuity of reward, $r$, transition, $P$, and policy function, $\pi$ in $L_p$ norm can be stated as below.
> 	\begin{align*}
> 	&(a)~ |r(x, u, \boldsymbol{\mu}_1, \boldsymbol{\nu}_1)-r(x, u, \boldsymbol{\mu}_2, \boldsymbol{\nu}_2)|_p\leq L_R^p[|\boldsymbol{\mu}_1-\boldsymbol{\mu}_2|_p + |\boldsymbol{\nu}_1-\boldsymbol{\nu}_2|_p] \\\\
> 	&(b)~ |P(x, u, \boldsymbol{\mu}_1, \boldsymbol{\nu}_1)-P(x, u, \boldsymbol{\mu}_2, \boldsymbol{\nu}_2)|_p\leq L_P^p[|\boldsymbol{\mu}_1-\boldsymbol{\mu}_2|_p + |\boldsymbol{\nu}_1-\boldsymbol{\nu}_2|_p] \\\\
> 	&(c)~|\pi(x,\boldsymbol{\mu}_1)-\pi(x,\boldsymbol{\mu}_2)|_p\leq L_Q^p|\boldsymbol{\mu}_1-\boldsymbol{\mu}_2|_p
> 	\end{align*}
>
> where $x, u$ are arbitrary state, and action; $\boldsymbol{\mu}_1, \boldsymbol{\mu}_2$ are arbitrary state distributions and $\boldsymbol{\nu}_1, \boldsymbol{\nu}_2$ are arbitrary action distributions. The terms $L_R^p, L_P^p$, and $L_Q^p$ are arbitrary positive reals and represent the Lipschitz constants in the $L_p$ norm. Consider the following relation for an arbitrary $K$-dimensional vector $X$.
> 	\begin{align}
> 	K^{-(1-\frac{1}{p})}|X|_1\overset{(a)}{\leq} |X|_p \overset{(b)}{\leq} |X|_1
> 	\end{align}
>
> Inequality (a) follows from Holder's inequality and (b) is an elementary relation between different norms. Using this, the Lipschitz continuities can be re-stated as follows.
> 		\begin{align*}
> 	&(a)~ |r(x, u, \boldsymbol{\mu}_1, \boldsymbol{\nu}_1)-r(x, u, \boldsymbol{\mu}_2, \boldsymbol{\nu}_2)|_1\leq L_R^p\lbrace|\boldsymbol{\mu}_1-\boldsymbol{\mu}_2|_1 + |\boldsymbol{\nu}_1-\boldsymbol{\nu}_2|_1\rbrace \\\\
> 	&(b)~ |P(x, u, \boldsymbol{\mu}_1, \boldsymbol{\nu}_1)-P(x, u, \boldsymbol{\mu}_2, \boldsymbol{\nu}_2)|_1\leq |\mathcal{X}|^{(1-\frac{1}{p})} L_P^p\lbrace|\boldsymbol{\mu}_1-\boldsymbol{\mu}_2|_1 + |\boldsymbol{\nu}_1-\boldsymbol{\nu}_2|_1\rbrace \\\\
> 	&(c)~|\pi(x,\boldsymbol{\mu}_1)-\pi(x,\boldsymbol{\mu}_2)|_1\leq |\mathcal{U}|^{(1-\frac{1}{p})} L_Q^p|\boldsymbol{\mu}_1-\boldsymbol{\mu}_2|_1
> 	\end{align*}
>
> Therefore, a preliminary bound on the approximation error for $L_p$ norm, $p\geq 1$ can be obtained by replacing $L_R, L_P, L_Q$ by $L_R^p, |\mathcal{X}|^{(1-\frac{1}{p})}L_P^p$, and $|\mathcal{U}|^{(1-\frac{1}{p})}L_Q^p$ respectively in all of our theorems. However, a detailed investigation is the subject of future work, and is not the focus of this paper, since the overall results are stated in the $L_1$ error.
>
> $\textbf{Q3: On the allowable range of}$ ~$\gamma$~ $\textbf{and the importance of}~L_Q:$
>
> Theorem 1, 2, and 3 in our paper clearly dictate that the approximation error can be bounded as long as $\gamma S_P<1$. Outside of this range, our results do not provide any guarantees. Therefore, we agree that our results indicate $\gamma$ has to be small for the approximation bound to be finite.
>
> We note that the term $L_Q$ is not an artifact of our proof. It did not appear in the results of (Gast and Gaujal 2011) probably because of its fundamentally different problem setting.
>
> $\textbf{Q4: What is known to the learner?}$
>
> We apologize for the confusion. In the revised manuscript (Sec. 7), we have clarified that the mean-field learner is aware of the transition function, $P$, but may not be aware of the reward function, $r$. It is to be mentioned that the availability of the knowledge of $P$ is a necessary assumption in the mean-field framework because the exact state distribution of an infinite agent system cannot be physically observed but must be iteratively obtained via the mean-field update equation (Eq. (8) in the manuscript).
>
> $\textbf{Q5: Is Algorithm 1 specific to the current problem or a reminder of an existing algorithm?}$
>
> Algorithm 1 is a process to solve the Mean-Field Control (MFC) problem. It is an adaptation of the Natural Policy Gradient (NPG) based algorithm of (Liu et al., 2020) into the mean-field setup.
>
> $\textbf{Q6: In Algorithm 2, how does line 3 work?}$
>
> As clarified in response to Q4, the availability of the knowledge of $P$ is presumed in the mean-field framework.
>
> $\textbf{Q7: On equation (19) and (20):}$
>
> In the revised manuscript, we have made appropriate changes in the notations.

---

> ### Author Response · Authors · 2022-08-02
> **Response to Reviewer 84q8: Part III**
>
> $\textbf{Q8: Elaboration of sample complexity and the use of generative model:}$
>
> Lemma 1 (which states the sample complexity result) is essentially an adaptation of Theorem 4.9 of (Liu et al., 2020) to our mean-field setting. Following that result, the sample complexity bound can be elaborated as $\mathcal{O}\left((1-\gamma)^{-6}\epsilon^{-3}\right)$. We do use a generative model. It is evident from line 1 of the subroutine $\mathrm{Update}$ of Algorithm 3 (which is a part of Algorithm 1) where we conditionally sample the next state of the mean-field representative given its current (state, action) pair and the state, action distributions of the whole population.
>
> (Liu et al., 2020) An improved analysis of (variance-reduced) policy gradient and natural policy gradient methods. Advances in Neural Information Processing Systems, 33:7624–7636, 2020.
>
> $\textbf{Q9: On the literature of non-uniform models:}$
>
> Thank you for suggesting the paper. We hope that it will be helpful in our future works.

---

### Decision · Action_Editors · 2022-09-06

**Recommendation:** Accept with minor revision

**Comment:**

This is a mostly theoretical paper about cooperative multiagent reinforcement learning (MARL) with N agents. There are several contribution of the paper: a bound on the distance of reward between local policies (that can be implemented by agents decentrally) compared to the globally optimal policy, an algorithm that computes the approximate optimal mean-field sequence of policies (upon which the local policies are based), and the formal relationships between n-player MARL case and mean-field control.

Every reviewer stated something about that clarity (either the quality of writing or the intuitive flow of the proofs / logic). In addition, I feel the presentation is particularly well-structured. This is especially difficult for a densely technical paper like this one. The contributions being made are clear and important to the growing field of MARL. There were several technical discussions from points brought up by reviewers, that the authors have included detailed responses in the submission system. If they have not already, I encourage them to integrate their responses into the paper text for improved clarity on these points, as readers are likely to have similar questions.